



# Evaluating the physical and biogeochemical state of the global ocean component of UKESM1 in CMIP6 Historical simulations

Andrew Yool[1], Julien Palmiéri[1], Colin G. Jones[2], Lee de Mora[3], Till Kuhlbrodt[4], Ekatarina E. Popova[1], A.J. George Nurser[1], Joel Hirschi[1], Adam T. Blaker[1], Andrew C. Coward[1], Edward W. Blockley[5], and Alistair A. Sellar[5]

[1]National Oceanography Centre, European Way, Southampton SO14 3ZH, UK
[2]National Centre for Atmospheric Science, University of Leeds, Leeds LS2 9JT, UK
[3]Plymouth Marine Laboratory, Prospect Place, Plymouth PL1 3DH, UK
[4]National Centre for Atmospheric Science, University of Reading, Earley Gate, Reading RG6 6BB, UK
[5]Met Office, FitzRoy Road, Exeter, Devon EX1 3PB, UK

**Correspondence:** Andrew Yool (axy@noc.ac.uk)

**Abstract.** The ocean plays a key role in modulating the climate of the Earth system (ES). At the present time it is also a major sink both for the carbon dioxide ($CO_2$) released by human activities as well as for the excess heat driven by the resulting atmospheric greenhouse effect. Understanding the ocean's role in these processes is critical for model projections of future change and its potential impacts on human societies. A necessary first step in assessing the credibility of such future
5   projections is an evaluation of their performance against the present state of the ocean. Here we use a range of observational properties to validate the physical and biogeochemical performance of the ocean component of UKESM1, a new Earth system (ESM) for CMIP6 built upon the HadGEM3 physical climate model. Analysis focuses on the realism of the ocean's physical state and circulation, its key elemental cycles, and its marine productivity. UKESM1 generally performs well across a broad spectrum of properties, but it exhibits a number of notable biases. Physically, these include a global warm bias inherited from
10   model spin-up, excess northern sea-ice but insufficient southern sea-ice, and sluggish interior circulation. Biogeochemical biases found include shallow remineralisation of sinking organic matter, excessive iron stress in regions such as the Equatorial Pacific, and generally lower surface alkalinity that results in decreased surface and interior dissolved inorganic carbon (DIC) concentrations. The mechanisms driving these biases are explored to identify consequences for the behaviour of UKESM1 under future climate scenarios, and avenues for model improvement. Finally, across key biogeochemical properties, UKESM1
15   improves in performance relative to its CMIP5 precursor, and compares favourably to fellow members of the CMIP6 ensemble.

**Key points**

- Physical and biogeochemical evaluation of the ocean component of the UKESM1 model

- Identification and investigation of present-day biases in major properties

- UKESM1 improves on CMIP5 predecessor and compares well with CMIP6 ensemble







# 1 Introduction

The climate dynamics of the Earth system are a product in large part of the two interacting geophysical fluids at the planet's
surface: the atmosphere and the ocean. Both are reservoirs for heat and the greenhouse gas carbon dioxide ($CO_2$), one of
several climatically-relevant chemical constituents. Because of the high specific heat capacity of water, as well as the chemical
buffering capacity of seawater, the ocean stores the majority of the Earth system's active reserves of both. Over the past few
centuries, the atmospheric concentration of $CO_2$ has risen exponentially from its quasi-stable interglacial background of around
278 ppm to more than 400 ppm. This growth is largely in response to the release of $CO_2$ through anthropogenic processes such
as fossil fuel combustion, land clearance and cement production. This change in the chemical composition of the atmosphere
has also altered its radiative transfer properties toward retaining a greater fraction of outgoing long-wave radiation, resulting in
atmospheric warming and change to the climate of the Earth system. Further, for the reasons identified above, the ocean is the
destination for the majority of these anthropogenic perturbations in both heat and carbon dioxide (e.g. Archer, 2005; Kuhlbrodt
et al., subm.).

Leaving aside the relatively static inventory within the geosphere, the Earth's carbon cycle partitions this element dynami-
cally between atmosphere, ocean and land systems, including the living systems of the marine and terrestrial biosphere. While
ongoing climate change is driven in the first instance by change in carbon (as $CO_2$) in the atmosphere, this reservoir represents
only approximately 1.4% of the total (pre-industrial) dynamic pool (Ciais et al., 2013), compared with 6.0% for land systems
(excluding permafrost) and 92.6% for ocean systems (excluding seafloor sediments). This dominance of the ocean reflects the
solubility of inorganic carbon in seawater, and ultimately the majority fraction of these anthropogenic emissions is expected to
be absorbed into the ocean (Archer, 2005). However, the magnitude of this, as well as the rate at which it occurs is dependent
upon a raft of physico-chemical and biological processes, including surface solubility, deep ocean ventilation and circulation,
and biological uptake and deep sequestration via sinking biogenic particles. Representing this uptake within an ESM requires
realistic performance across many aspects of its simulated ocean state, both physical and biogeochemical, and both surface and
interior.

The situation is similar for heat, with observations over recent decades showing a clear upward trend in ocean heat content
since the 1960s at the earliest, and accelerating since the 1990s (Levitus et al., 2012; Cheng et al., 2017). Approximately 90%
of the anthropogenic imbalance in the Earth's heat content is stored within the ocean (Meyssignac et al., 2019). Consequently,
and similarly to carbon, simulating this important property requires ESMs to accurately represent a broad range of physical
phenomena, such as ocean circulation and mixing that distribute heat, as well as sea-ice that caps its exchange and affects
albedo.

This manuscript is concerned with the realism of the ocean component of UKESM1, during CMIP6 Historical period simu-
lations (1850-2014; Eyring et al., 2016. It has three primary goals:

- First, to evaluate the performance of UKESM1 against observational metrics, and identify biases in physical and biogeo-
  chemical properties.

- Second, to identify the first-order causes of biases found, to elucidate where modelled processes may be less realistic.





    – Third, to identify specific weaknesses and avenues for future model improvement or development.

Model performance is evaluated across a broad range of properties to identify biases, with analysis focusing on the near-present period of 2000-2009 because of the greater availability of observational data in recent decades. Overall, the manuscript aims
to facilitate subsequent more in-depth analyses of the model by identifying ocean states or processes where its representation is weaker. A summary analysis across all of UKESM1's components can be found in Sellar et al. (2019).

    The manuscript is structured as follows. A brief introduction to UKESM1 is presented, with an emphasis on its ocean components, followed by outlines of the model simulations used and the observational datasets selected for their evaluation. Results are then presented for the physical ocean, sea-ice and marine biogeochemistry components, with surface and interior
bulk properties, dynamical and biogeochemical processes, and time-series examined. Discussion is focused on the major biases identified, proposals for reducing these in future model revisions, and an evaluation of UKESM1 in the context of peer (and precursor) CMIP models.



## 2 Methods

### 2.1 Earth system model

This study utilises UKESM1, a new state-of-the-art model built to describe the coupled physical and biogeochemical dynamics of the Earth system including its atmosphere, ocean and land systems. UKESM1 uses the Hadley Centre Global Environment Model version 3 Global Coupled (GC) version 3.1 configuration, HadGEM3 GC3.1 (Williams et al., 2017; Kuhlbrodt et al., 2018), as its core physical climate model. This is then extended through the addition of interactive stratospheric–tropospheric trace gas chemistry, land biogeochemistry and ecosystem dynamics, and ocean biogeochemistry. In addition to the internal dy-

namics of these components, the resulting ESM includes couplings between them to represent potential feedback processes or interactions that may impact the time-evolution of the modelled climate. Sellar et al. (2019) provides an overview of UKESM1, including its development and tuning, while Yool et al. (2020) describes the spin-up of its pre-industrial control (piControl) state ahead of Historical period (1850-2014) simulations.

Supplementary Figure S1 shows a schematic overview of the constituent models of UKESM1. In outline, UKESM1 is

comprised of distinct atmosphere and land blocks that are coupled closely together, alongside an ocean and sea-ice block that is linked to the atmosphere through an explicit coupler module, OASIS3-MCT_3.0 (Valcke, 2013; Craig et al., 2017). All three major ES components – atmosphere, land and ocean – are themselves built from submodels that separately represent domains such as physical dynamics, biogeochemistry and ecosystem dynamics.

The physical dynamics of the atmosphere of UKESM1 are represented by GA7.1 (Mulcahy et al., 2018; Walters et al., 2019),

which includes processes such as mass transport, radiative transfer, thermodynamics and the water cycle. The UK Chemistry and Aerosols model (UKCA; Morgenstern et al., 2009; O'Connor et al., 2014) is coupled to GA7.1, and includes stratospheric and tropospheric chemistry together with separate aerosol (Mann et al., 2010) and dust schemes (Woodward, 2011). UKESM1 adds several couplings that are absent in GA7.1, including natural emissions of monoterpenes, dimethyl sulphide (DMS) and primary marine organic aerosols (PMOA), all of which are calculated dynamically from land and ocean components and which

permit additional climate feedbacks. The atmosphere in UKESM1 also serves as a conduit for mineral dust, transferring this from bare soil on land into the ocean where it can fuel biological production and $CO_2$ uptake. Mulcahy et al. (2018), Sellar et al. (2019), Archibald et al. (2020) and Mulcahy et al. (subm.) provide further details of the atmospheric chemistry and aerosol schemes in UKESM1.

Physics and biogeochemistry on land in UKESM1 is represented by the Joint UK Land Environment Simulator (JULES;

Best et al., 2011; Clark et al., 2011). This is closely coupled to the Top-down Representation of Interactive Foliage and Flora Including Dynamics model (TRIFFID; Cox, 2001; Jones et al., 2011), which represents plant and soil dynamics on land. TRIF-FID developments new to CMIP6 include updated plant parameterisations (Kattge et al., 2011), increased plant functional types (Harper et al., 2016), the production of volatile organic compounds (Pacifico et al., 2015), and nitrogen limitation of terrestrial primary production and carbon uptake (Wiltshire et al., subm.). TRIFFID represents land-use by agriculture by reserving grid

cell time-varying fractions for occupation by crops and pasture. For further details of UKESM1's land component, please refer to Sellar et al. (2019).





The physical ocean component in UKESM1 makes use of the Nucleus for European Modelling of the Ocean framework (NEMO; Madec et al., 2016) This is comprised of an ocean general circulation model, Océan PArallélisé version 9 (OPA9; Madec et al., 1998; Madec, 2008), and is coupled here to a separate sea-ice model, the Los Alamos Sea Ice Model ver-
sion 5.1.2 (CICE; Hunke et al., 2015). OPA9 is a primitive equation model of ocean dynamics, and is used within UKESM1 at a horizontal resolution of approximately 1° on a tripolar grid (Madec and Imbard, 1996) with enhanced equatorial resolution (the extended ORCA1 grid, or eORCA1). This shared configuration of NEMO, dubbed "shaconemo", is used by a number of European research groups, and many of its grid resolution-dependent settings are aligned with these other ESMs (NEMO v3.6_stable; available from `http://forge.ipsl.jussieu.fr/shaconemo`). Some other parameterisations (typically
resolution-independent) are drawn from the GO6 configuration of NEMO developed in the UK (Storkey et al., 2018). More complete descriptions of the NEMO and CICE configurations used in UKESM1 (GO6, GSI8), including details of its sensitivity and resulting tuning, can be found in Storkey et al. (2018), Ridley et al. (2018) and Kuhlbrodt et al. (2018), while Kuhlbrodt et al. (subm.) investigates ocean heat uptake.

Marine biogeochemistry in UKESM1 is represented by the Model of Ecosystem Dynamics, nutrient Utilisation, Sequestra-
tion and Acidification (MEDUSA-2.1). MEDUSA-2.1 is "intermediate complexity" with a double size-class ecosystem that represents phytoplankton, zooplankton and particulate detrital pools, and which explicitly includes the biogeochemical cycles of nitrogen, silicon and iron nutrients as well as the cycles of carbon, alkalinity and oxygen. During its inclusion within UKESM1, a number of changes were introduced from its earlier predecessor model, MEDUSA-2, described in Yool et al. (2013), and the version used here is identified as MEDUSA-2.1, to distinguish it. These changes include: 1. adoption of the
MOCSY-2.0 carbonate chemistry scheme of Orr and Epitalon (2015); 2. inclusion of surface seawater dimethyl sulphide (DMS) concentration using the empirical submodel of (Anderson et al., 2001); 3. code improvements including variable volume (VVL) and the XML Input-Output Server (XIOS) (Meurdesoif, 2013). A more complete description of MEDUSA-2.1 can be found in the Appendix A.

## 2.2 Model simulations

This study utilises simulations of the UKESM1 model performed as part of the 6th phase of the Coupled Model Intercomparison Project (CMIP6). Model output is taken from the piControl and Historical simulations of CMIP6, and from an ensemble of 9 members, consistent with Sellar et al. (2019). Each ensemble member represents a branch at a different time point from the piControl, after which the new simulation experiences time-varying changes in atmospheric and land-use properties characteristic of the Historical period from start-1850 to end-2014. Ensemble branch points were chosen selectively to span the
variability in the model's multi-decadal behaviour (Sellar et al., 2019). To achieve this the model's behaviour across two major ocean modes was sampled: the Atlantic Multi-decadal Oscillation (AMO; Kerr, 2000), and the Inter-decadal Pacific Oscillation (IPO; Zhang et al., 1997; Power et al., 1999). Supplementary Table S1 lists the local run IDs of the simulations comprising the ensemble, together with their branch times from the piControl. The mean of this 9 member ensemble is used throughout the following analysis, except where stated otherwise.





## 2.3 Model evaluation

Model analysis in this study is focused on a subset of ocean properties. More complete evaluations of other UKESM1 components can be found in the dedicated studies of Mulcahy et al. (2018) and Mulcahy et al. (subm.) (aerosols), Archibald et al. (2020) (atmospheric chemistry) and Andrews et al. (2019) (radiative forcing, feedbacks and climate sensitivity). Sellar et al. (2019) provides a summary overview of the full model.

Evaluation uses the period 2000-2009 of the CMIP6 Historical simulation and compares to corresponding periods of observational data. Some evaluated properties are not as comprehensively sampled, but the same time period is likely to be representative of the ocean's state so we use this for consistency. The results shown make use of monthly climatologies of both model output and observational data (where available) for this period.

The specific observational datasets used for evaluation are as follows:

– World Ocean Atlas 2013, for ocean physics (interior; Locarnini et al., 2013; Zweng et al., 2013) and biogeochemistry (interior, surface; Garcia et al., 2014; Garcia et al., 2014) fields

– Hadley Centre Sea Ice and Sea Surface Temperature (HadISST.2.2; Titchner and Rayner, 2014), for ocean SST and sea-ice fields

– National Sea Ice Data Centre for sea-ice thickness (Stroeve and Meier, 2016) and sea-ice index (Fetterer et al., 2017)

– Estimating the Circulation and Climate of the Ocean (ECCO) V4r4 (Forget et al., 2015; Fukumori et al., 2019), for ocean circulation

– Smeed et al. (2018) for RAPID-MOCHA time-series measurements of the Atlantic meridional overturning circulation at 26°N

– SeaWiFS (O'Reilly et al., 1998), for surface ocean chlorophyll concentration

– Oregon State University Ocean Productivity group, for VGPM (Behrenfeld and Falkowski, 1997), Eppley-VGPM (Carr et al., 2006) and CbPM (Westberry et al., 2008) vertically-integrated primary production

– Rödenbeck et al. (2013) for observationally-derived global air-sea $CO_2$ flux and surface $pCO_2$

– Lana et al. (2011) for surface dimethyl sulfide (DMS) concentrations

– Global Ocean Data Analysis Project v1.1 (Key et al., 2004) and v2 (Key et al., 2015; Lauvset et al., 2016), for interior and surface carbonate biogeochemistry

– Moriarty and O'Brien (2013) for the COPEPOD dataset of gridded zooplankton biomass

Links to these datasets are given in Appendix D.





## 3 Results

### 3.1 Surface physical ocean

Figure 1 shows observed (HadISST; Titchner and Rayner, 2014) and simulated global-scale sea surface temperature (SST) for summer and winter in both hemispheres, together with (model - observed) patterns of difference. The model reproduces the main observed features, including latitudinal and seasonal gradients, upwelling regimes and major fronts. A number of biases are also evident, including warm biases up to 4°C in upwelling regimes (especially the equatorial Pacific), a general warm bias in the Southern Ocean, cool biases of up to -2°C throughout the subtropics, and a marked cold bias in the North Atlantic of

greater than -4°C. The former Pacific biases occur in December-January-February (DJF) when tropical atmospheric convection is primarily over the western Pacific warm pool, and the east-west pressure gradient is seasonally at a maximum. This gradient drives east-west wind stress and equatorial Ekman-induced upwelling, and a poor representation of this in UKESM1 likely leads to reduced upwelling and the warm SST bias. A warm bias close to the North American coastline and strong cold bias in the western North Atlantic occur due to resolution-dependent errors where the Gulf Stream separates too far north and then

extends too zonally across the North Atlantic (Marzocchi et al., 2015; Hirschi et al., 2020). % citepmarzocchi2015, and Similar but less marked biases occur in the Pacific in association with the Kuroshio Current. In general, surface temperature biases in the model have strong latitudinal patterns associated with major currents and patterns of upwelling and downwelling, and are persistent across the seasons. To illustrate the full seasonal cycle, Supplementary Figure S3 shows Hovmöller diagrams of latitudinal mean observed and simulated SST.

Remaining with the surface ocean but moving to high latitude regions, Figure 2 shows the observed and simulated sea-ice concentrations at the seasonal maxima, March in the Arctic and September in the Antarctic (HadISST; Titchner and Rayner, 2014). In general terms, the model reproduces the observed northern hemisphere sea-ice patterns, with complete ice cover in the main Arctic basin, Baffin Bay down to Davis Strait, and Hudson Bay, cover on the eastern margins of Newfoundland and Greenland, and bounding the Barents Sea. In the Arctic, simulated maximum sea-ice area is $15.3 \times 10^6$ km$^2$, compared with

an observational maximum of $13.9 \times 10^6$ km$^2$. This relationship is reversed in the Antarctic, with a simulated maximum of $11.8 \times 10^6$ km$^2$ compared to $16.3 \times 10^6$ km$^2$ observed. As the bottom row of Figure 2 shows, this general pattern of excess sea-ice in the Arctic and a deficit around Antarctica generally persists seasonally, with a modelled Arctic minimum of 8.7 compared to $4.7 \times 10^6$ km$^2$ observed, and a model Antarctic minimum of 2.7 compared to $2.6 \times 10^6$ km$^2$ observed. Modelled Arctic sea-ice also reaches its seasonal minimum slightly earlier than observed, in August rather than September. In the Arctic,

sea-ice is typically multi-year, and this positive bias in modelled area is accompanied by excessively thick sea-ice. Thicknesses are up to 5 m in the simulated "dome" of sea-ice over the north pole, compared to flatter observational estimates that are closer to 3 m (Supplementary Figure S5; Stroeve and Meier, 2016).

In response to ongoing climate change, Arctic sea-ice shows one of the most pronounced trends within the Earth system over recent decades (Brennan et al., 2020). Figure 3 shows simulated Arctic and Antarctic sea-ice extent over the full Historical

period (1850–2014), together with observational estimates (HadISST, Titchner and Rayner, 2014; NSIDC, Fetterer et al., 2017) and for recent decades. Much as with sea-ice extent itself, UKESM1 performs best in the Arctic, with similar negative trends





since 1980. In the Antarctic, however, the discrepancy in seasonal extent already noted is exacerbated by a negative trend in maximum sea-ice extent in UKESM1 opposite to the rising trend actually observed (although this observed trend may be reversing; Parkinson, 2019).

The Earth's ocean and atmosphere interact principally at their interface, but turbulent mixing of the ocean ventilates its upper layer with both physical and biogeochemical consequences. Figure 4 shows the observed and modelled thickness of this mixed layer, together with (model - observed) patterns of difference. Mixed layer thickness (MLD) is estimated here as the depth at which the vertical profile of potential temperature changes by 0.5°C from its 5 m value (WOA; Locarnini et al., 2013). Again, the model reproduces the main features of the ocean, including strong seasonality at high latitudes, deep mixed layers

(> 100 m) throughout the year in the Southern Ocean (away from sea-ice), and shallow mixed layers (< 50 m) in equatorial upwelling regions. When and where the mixed layer is shallow, the model tends to exaggerate this with even shallower mixed layers, most noticeably during the summer at temperate latitudes. At subpolar latitudes, in the Southern, Atlantic and Pacific oceans, deep mixing in the winter is more pronounced in the model, with larger areas experiencing mixing to deeper than 500 m. These model biases towards both shallower and deeper mixed layer depths are more clearly visible in Figure 5, which

shows the frequency at which different mixed layer depths occur seasonally. While median frequencies are similar between the model and those observation-derived, modelled summer and winter distributions can be seen to be shifted shallow and deep respectively.

## 3.2    Interior physical ocean

Switching to the ocean interior, Figures 6 and 7 respectively illustrate zonally-averaged depth profiles of temperature and

salinity along so-called "thermohaline transects" of the Atlantic, Southern and Pacific oceans, for both UKESM1 and obser-vations (Locarnini et al., 2013; Zweng et al., 2013). These transects track southward down the Atlantic into the Southern, before reversing direction to travel northward from the Southern into the Pacific, with the aim of broadly following watermass properties from young, freshly-ventilated North Atlantic Deep Water (NADW) through to much older North Pacific waters. For the purposes of this transect, the Arctic Ocean is considered a northern extension of the Atlantic, while the Indian Ocean

– west of the Malay Archipelago, and including its sector of the Southern Ocean – is entirely omitted from consideration. In both cases, observed and modelled interior properties are shown, together with a difference plot to highlight biases.

For ocean temperature, while there are spots of cooler biases in the upper ocean (< 1000 m), temperature is generally positively biased in the upper 3000 m. This is more pronounced in the Atlantic basin, in particular at tropical latitudes, where midwater (100–1000 m) biases up to 4°C are found in the model. Comparable Pacific biases are much lower, and tropical

latitudes instead show a cold bias in the upper 500 m. At depth, both basins show negative biases, which again are more pronounced in the Atlantic. Southern Ocean temperatures exhibit small postive biases, most clearly in the Atlantic sector, although these switch sign at depth into the Atlantic proper as already mentioned. Patterns of ocean salinity broadly mirror those of temperature in the Atlantic basin, with corresponding positive biases in the upper 3000 m, and negative biases below. The model's Pacific basin is more uniformly fresh in the upper 1000 m, with smaller positive biases beneath, and negligible

biases below 3000 m. Overall, temperature and salinity patterns indicate that the Atlantic is a warmer, more evaporative





basin in the model, with its most positive upper ocean biases located there, as well as its largest negative biases in the deep ocean. Supplementary Figure S6 shows the corresponding patterns in potential density anomaly ($\sigma_\theta$; referenced to atmospheric pressure). These show the model ocean, particularly the Pacific basin, to be more stratified vertically compared to observations, with generally lower density surface waters ($< 1000$ m) overlying more dense deep waters.

This pattern of biases in the zonal sections above indicates differences in the balance of interior watermasses in UKESM1 compared to that of the real ocean. Observationally, zonally-averaged North Atlantic circulation below 1000 m is dominated by the transports associated with North Atlantic Deep Water (NADW) and the Antarctic Bottom Water (AABW). NADW is produced by the the subduction of cool, salty water at subpolar latitudes in north of the basin, and its southward-moving cell overlies a denser cell of Antarctic Bottom Water (AABW) travelling northward from its production in the Southern Ocean.

To illustrate this, the upper panel of Figure 8 shows a reconstruction of the global streamfunction of the ocean's meridional overturning circulation (MOC), produced by the Estimating the Circulation and Climate of the Ocean consortium (ECCO; Forget et al., 2015; Fukumori et al., 2019). This is an ocean reanalysis product in which the MOC is a result of a model simulation that has been constrained with observations (for a more complete overview, see Jackson et al., 2019). In this, the upper positive (clockwise) overturning cell extends its influence below 2000 m (in red; driven by circulation in the North

Atlantic), overlying the negative overturning cell (in blue) of AABW. The lower panel of Figure 8 shows the corresponding MOC in UKESM1. In general, this follows the pattern shown in the ECCO reanalysis, although with a slightly stronger maximum MOC at 40°N, and a weaker AABW cell northward of the Antarctic Circumpolar Current (ACC). We note that the southernmost part of the overturning associated with AABW is stronger in UKESM1 than in ECCO (around 6 Sv against 4 Sv), suggesting that sinking around Antarctica is stronger in UKESM1. Stronger sinking around Antarctica, combined with a

slightly weaker NADW, is consistent with the colder and fresher conditions shown for the deep ocean (particularly the Atlantic) in Figures 6 and 7, as well as biases in biogeochemical fields (see later).

    While Figure 8 shows a time- and zonally-averaged state of the MOC, ocean circulation exhibits significant variability (Majewski et al., 2009; Smeed et al., 2018). Annual mean observation-based estimates of the Atlantic MOC (AMOC) from the RAPID-MOCHA array at 26.5°N range from 14.6–19.3 Sv between 2004–2016 (Smeed et al., 2018). In the Southern Ocean,

Drake Passage, the channel between the Antarctic Peninsula and South America, focuses the ACC that rings Antarctica, and from intermittent sampling has a transport estimated at $173 \pm 11$ Sv (Donohoe et al., 2016). Figure 9 shows time-series of both of these major transports across the full Historical period, for all 9 ensemble members. UKESM1's pre-industrial AMOC is typically lower than that found by RAPID-MOCHA (Yool et al., 2020) (consistent with the spatial displacement mentioned previously), but strengthens by approximately 3 Sv from 1850 to a maximum of around 17 Sv by the 1990s. This increase

in AMOC strength, which ends in UKESM1 around 2000, is almost certainly causally linked to temporal trends in negative radiative forcing driven by anthropogenic aerosol emissions in the northern hemisphere over this period (Menary et al., 2020). Increases in these, driven by industrial activity, cool the north relative to the south, change the inter-hemispheric thermal gradient, and result in increasing AMOC strength in response. Although good observational data is absent prior to the construction of the RAPID-MOCHA array, this rise in AMOC strength is consistent with model reanalysis over this period (Jackson et al.,

2016), although possibly overestimated in CMIP6 models such as UKESM1 (Menary et al., 2020). The subsequent decline





during first decades of the 21st century matches that found by RAPID-MOCHA (Smeed et al., 2018) and reanalysis (Jackson et al., 2016). The modelled AMOC increase in UKESM1 is absent in the parallel segments of the piControl simulation that do not experience these anthropogenic changes (see the linear trends in Table 1).

Time-averaged over the Historical period ($\approx$ 150 Sv), Drake Passage transport in UKESM1 is lower than that estimated (Donohoe et al., 2016), although across the full ensemble and its long-period variability, the model intermittently reaches the range observed (Figure 9). Throughout the Historical period the ensemble exhibits considerable multi-decadal to centennial scale variability in modelled ACC strength (135–173 Sv). Unlike AMOC strength, where the ensemble shows a clear trend that all members follow, ACC strength is much less aligned across the ensemble, most clearly in the period 1850–1930. Between 1930–1980, however, the ensemble spread is reduced and most ensemble members exhibit a weak ACC. However, following this point most strengthen notably, recovering from this earlier minimum to reach higher values more consistent with the recent observations. The increase in ACC strength post-1970 is consistent with development of the Antarctic ozone hole and strengthened westerlies over the Southern ocean which then drives a stronger ACC (e.g. Li et al., 2016). Nonetheless, as Figure 9 shows, two of the nine members do not exhibit this minimum around 1970, suggesting that while a forced climate driver may be operating on ACC strength, it cannot completely override internal variability in the Southern Ocean.

### 3.3 Surface nutrient biogeochemistry

Figures 10 to 16 present model-observation intercomparisons for a range of key surface biogeochemical properties, showing seasonal geographical fields and zonal Hovmöller diagrams (where possible). Table 3 lists global and regional means for the same properties on an annual mean timescale. To summarise across these properties, Supplementary Figure S10 additionally shows seasonal and regional Taylor diagrams.

In terms of surface concentrations of the macronutrients that regulate biological productivity in the ocean, UKESM1 shows some shared and some divergent biases. For dissolved inorganic nitrogen (DIN; Figure 10), while the major, circulation-driven features occur (i.e. subtropical gyre lows, upwelling highs), the model is typically biased positive, with excess nutrients most obvious in the tropical Pacific and in the Arctic Ocean (see also Supplementary Figure S19). Globally, the model's mean is 7.8 compared to an observational mean of 5.2 mmol m$^{-3}$ (+48%). However, in regions such as the North Atlantic, the model is biased negative with winter maximum concentrations much lower ($\approx$ 5 vs. $\approx$ 10 mmol m$^{-3}$) in this important productive region. The North Pacific, by contrast, exhibits the year-round high nutrient concentrations that characterise this region (12 vs. 10 mmol m$^{-3}$). However, the spatial distribution of North Pacific DIN, particularly around the Bering Straits, biases inflow concentration to the Arctic Ocean and is responsible for the excess concentration in this region.

In MEDUSA, silicic acid is a key limiting factor for the growth of the model's large phytoplankton, the diatoms. As Figure 11) shows, away from the Southern Ocean where it is strongly biased positive ($\approx$ 63 vs. $\approx$ 32 mmol m$^{-3}$; Table 2), the model is typically biased negative. Globally, the model's mean is 10.1 compared to an observational mean of 7.5 mmol m$^{-3}$ (+50%). While silicic acid concentrations are generally low throughout the tropical and subtropical ocean (maxima < 20 mmol m$^{-3}$), modelled concentrations are much more depleted throughout the year (maxima < 5 mmol m$^{-3}$). In the North Pacific, un-





like with DIN, seasonal maximum silicic acid concentrations are signficantly lower than observed in this region (4.5 vs.
21.3 mmol m$^{-3}$).

Alongside nitrogen and silicon (the latter for diatoms only), phytoplankton productivity in MEDUSA is additionally lim-
ited by the micronutrient, iron. An important source of iron to the ocean is via deposition of aeolian dust that has been lifted
from dessicated land surfaces and transported by winds (Tagliabue et al., 2017; Kok et al., 2018). MEDUSA represents this
source of iron to the ocean, and in UKESM1 this flux of dust is driven by dynamic land-atmosphere interactions (Woodward,
2011). Figure 12 compares the simulated flux of iron from dust with the observationally-derived dataset of Mahowald (2005).
Following Yool et al. (2013), dust is scaled in UKESM1 such that total iron added to the ocean by deposited dust is approx-
imately 2.6 Gmol Fe y$^{-1}$ (excluding the Mediterranean Sea), and the Mahowald (2005) panel is similarly scaled. In general,
UKESM1 exhibits similar spatial patterns to the observational product, including high deposition downwind of arid regions
such as the Sahara, and corresponding low deposition where airmasses do not intersect with land such as over the Southern
Ocean. However, several key areas of low deposition are more pronounced in the model, including the Southern Ocean, the Pe-
ruvian upwelling and the Equatorial Pacific. These regions are also those where excess DIN occurs, indicating that at least one
source for these biases may be excessively strong iron limitation on biological activity. To further illustrate this, Supplementary
Figure S18 shows the dominant nutrient limitation for both phytoplankton types. Notably, compared to other runs employing
MEDUSA (Yool et al., 2013), iron-stress is more pronounced in UKESM1, especially compared to nitrogen-stress, with the
Southern Ocean and almost the whole of the Pacific iron-limited for non-diatom phytoplankton, and diatom phytoplankton
iron-stressed across the Equatorial Pacific.

Switching to the biological community, Figure 13 presents surface chlorophyll, the main light-harvesting pigment used
by phytoplankton. Again, the model exhibits both postitive and negative biases relative to observations, but with a general
positive bias (0.26 vs. 0.22 mg chl m$^{-3}$). Most noticeably, modelled summer concentrations of chlorophyll in the Southern
Ocean are biased positive throughout the year, particularly so in the unproductive winter, when the model continues to simulate
moderate concentrations even at high latitudes (although winter observations are less reliable or absent). In part, the elevated
chlorophyll concentrations in UKESM1 are driven by the reduced extent of winter sea-ice in this hemisphere. At the equator,
the model is biased positive in the Pacific (0.25 vs. 0.18 mg chl m$^{-3}$), while strongly biased negative in the Atlantic (0.07 vs.
0.36 mg chl m$^{-3}$). Meanwhile, in the subtropical gyres, the model simulates lower concentrations than observed throughout,
particularly in the Atlantic Ocean, whereas the lowest observed concentrations occur in the southern Pacific subtropics. At high
northern latitudes, maximum chlorophyll concentrations are typically slightly lower than those observed, although, much as in
the southern hemisphere, moderate winter concentrations extend much further poleward than observed.

Figure 14 presents the corresponding distributions of net primary production, the process driving consumption of surface
nutrients, biological uptake of dissolved $CO_2$, and the ultimate source of organic matter for the ocean's food web. The ob-
servations shown here are the simple mean of three observation-driven estimates of productivity models: VGPM (Behrenfeld
and Falkowski, 1997); Eppley-VGPM (Carr et al., 2006); and CbPM (Westberry et al., 2008). Generally, although with some
of the same model biases already noted, simulated patterns clearly replicate those observed. Integrated globally, modelled
productivity across the UKESM1 ensemble averages 44.3 Pg C y$^{-1}$, compared with an average of 39.5 Pg C y$^{-1}$ estimated





by the three models. Regionally, the clearest bias lies, again, in the Southern Ocean, where modelled productivity is both
greater and geographically more extensive, with a large summer bloom that extends further south towards Antarctica (0.31 vs.
0.10 g C m$^{-2}$ d$^{-1}$). Another discrepancy lies in the tropics, where modelled productivity is more focused along the equator in
the Pacific, and with generally lower productivity in the subtropical gyres. Also, while productivity is focused in shelf regions
in both observations and the model, in the model it extends further into the open ocean than observed, where productivity
is generally restricted to a narrow band around the continents. Finally, in terms of seasonal extent, modelled productivity is
typically broader, with positive biases extending further polewards during winter in both hemispheres.

Supplementary Figure S8 shows the time-series of net primary production, and its main driver, DIN, across the Historical
period for all nine ensemble members. Earlier plots evaluated the geography and phenology of both fields in the early 21st
century, but this plot makes it clear that neither property is at equilibrium at this time. Global surface DIN shows a pronounced
rise (approximately 5%) from 1950 to around 2000, consistent across the ensemble, but by 2014 this increase has been en-
tirely reversed. Meanwhile, primary production has no clearly comparable 20th century trend, but from around 2000 declines
(approximately 2%). In terms of the main production regions, the North Atlantic and the Southern Ocean drive these global
signals, with production unsurprisingly lagging that of DIN (Supplementary Figure S8).

The critical role of primary production as the source of organic carbon (and chemical energy) on which marine ecology
runs means that the realism of its representation in models has consequences across marine biogeochemistry. To illustrate
this, Supplementary Figures S11 and S13 compare UKESM1 surface fields of a higher trophic level (mesozooplankton) and a
climatically-active biogenic gas (DMS) with observational estimates. As would be expected, both properties scale closely with
productivity, and share a number of the same geographical biases. While much of the ocean shows good model-observation
agreement, mesozooplankton biomass in the Southern Ocean is significantly elevated in both summer and winter compared
with Moriarty and O'Brien (2013)'s dataset, as well as more focused around the Antarctic Polar Front (Supplementary Figure
S11). The corresponding biomasses in both seasons in the northern hemisphere are better reproduced, although still with
biases, including lower North Pacific mesozooplankton, a region where their abundance has long been known to play a role in
seasonal dynamics (Steele and Henderson, 1992). Switching to DMS, the model actually shows pronounced negative biases in
the Southern Ocean, in contrast with other properties (Supplementary Figure S13). Elsewhere, regions of high concentration are
also typically more geographically confined in the model, with maximum values lower than those observed. The relatively good
general agreement with the observational (Lana et al., 2011) dataset in part relates to the tuning of the underlying (Anderson
et al., 2001) DMS model, although the divergence where observed concentrations are high, especially the Southern Ocean,
suggest this real-world property is a more complex function of primary production than modelled in UKESM1.

### 3.4 Surface carbon biogeochemistry

Figure 15 compares the annual mean surface concentrations of DIC and alkalinity, two key carbonate chemistry properties
that constrain the ocean's exchange of CO$_2$ with the atmosphere. In both cases, the model reproduces the spatial patterns well,
with main features such as elevated DIC at high latitudes, a strong Atlantic-Pacific alkalinity gradient, and generally lower
concentrations of both at lower latitudes. Globally, both model mean DIC and alkalinity are slightly biased negative compared





to observations, with implications for interior concentrations of DIC (see Section 3.5). Noticeable regional biases include positive biases for both properties in the Southern Ocean (particularly around Antarctica), and negative biases in alkalinity in
the North Atlantic and (especially) the North Pacific.

Critically linked to surface DIC and alkalinity, Figure 16 shows the observed and modelled patterns of air-sea exchange of $CO_2$. This is a key Earth system property, as its integrated magnitude modulates the accumulation of anthropogenic $CO_2$ in the atmosphere with its absorption by sinks such as the ocean and the land. The observational product used here fits a simple ocean mixed layer biogeochemistry scheme to observations of surface ocean $CO_2$ partial pressure, and then extrapoloates this
globally (Rödenbeck et al., 2013). Much as with its surface carbonate chemistry, the model reproduces the main features of air-sea $CO_2$ exchange, including zonal bands of ingassing and outgassing, pronounced equatorial outgassing in the Pacific, and strong seasonal ingassing at high latitudes in the northern hemisphere. However, the model also exhibits a number of biases in its regional and seasonal patterns of flux. While observations suggest that the Southern Ocean is a complex mix of summer ingassing and winter outgassing, the model is biased towards ingassing, with weaker and more geographically
limited outgassing in the southern winter. Further, though showing similar patterns to those observed, the model exaggerates seasonal ingassing in the northern hemisphere, particularly during late winter and spring at subtropical latitudes. Note that, again, the reliability of this observational product is lower in less sampled regimes, such as the SO and during winter. Overall, the ocean is a net sink for $CO_2$, with the model simulating total uptake of 2.05 Pg C $y^{-1}$ compared to an observational estimate of 1.60 Pg C $y^{-1}$ (although observational products differ on this quantity; see below). Supplementary Figure S12
shows corresponding plots of surface $pCO_2$, a function of surface DIC, alkalinity, temperature and salinity (Rödenbeck et al., 2013). Biases in these fields illuminate those in $CO_2$ flux, for instance much lower $pCO_2$ in the North Atlantic drives stronger uptake, while higher $pCO_2$ in the Southern Ocean damps down outgassing in this region.

To complement Figure 16's geographical snapshot, Figure 17 shows the time-series of $CO_2$ uptake across the Historical period for the UKESM1 ensemble, together with the observationally-derived estimate of Khatiwala et al. (2009). The plot
shows the varying rate in the rise of oceanic uptake of $CO_2$ across this period, with growth from the 1850s until the 1930s, followed by stalling growth until the 1950s, and finally strong continuous growth to the present-day. With some variability, particularly in the early decades, the ensemble tracks the observationally-estimated uptake, reproducing the same pace and features. The plot also shows UKESM1's piContol simulation to illustrate the magnitude and period of variability with constant background atmospheric $xCO_2$. Note that the observational estimate for the present-day here is more closely matched by the
model than the preceding dataset of Rödenbeck et al. (2013), although this is not unexpected given the large uncertainties involved in estimating this flux.

The air-sea flux is just the first stage of ocean storage of anthropogenic $CO_2$, and Figure 18 illustrates its fate once in the ocean interior. The upper row shows estimated and simulated vertically-integrated anthropogenic $CO_2$ for the 1990s (the normalised period for Key et al., 2004). In the case of the model, this is estimated by differencing Historical DIC fields
from each ensemble member with the corresponding DIC field from the same relative timepoint from the piControl. In broad outline, UKESM1 reproduces most of the geographical patterns of storage, with Southern Ocean uptake distributed into the southern sectors of the Atlantic, Pacific and (especially) Indian basins, maximum column inventories in the North Atlantic,





and much lower storage at low latitudes, in the North Pacific and around Antarctica. However, the modelled distributions of anthropogenic $CO_2$ also show some clear discrepancies with observational estimates. For instance, although exhibiting

high column inventories in the Greenland-Iceland-Norwegian Sea, the model ensemble does not simulate the corresponding high observationally-estimated concentrations off Newfoundland in the west of the Atlantic. More significantly, the pattern of anthropogenic $CO_2$ being transported southward at depth in the North Atlantic shows a strong east-west gradient that does not correspond with that observed. To investigate this further, Figure 20 shows observational and model sections across the Atlantic at 30°N for both anthropogenic $CO_2$ and CFC-11. The former is estimated from observations, while the latter is

measured directly. These show a general deficit in UKESM1 in tracer concentrations between approximately 1000 and 3000 m in depth west of the mid-Atlantic ridge. In the case of CFC-11, the model completely misses a distinctive watermass with high concentrations immediately adjacent to the coast of North America at approximately 1800 m. As already noted for the surface ocean in Section 3.1, grid resolution introduces errors into transport pathways, and UKESM1's poor representation of Deep Western Boundary Current (DWBC) return flow may be an interior example of similar limitations, coupled potentially to

discrepancies in patterns in convection and deep mixing in the vicinity of the Labrador Sea (e.g. Handmann et al., 2018).

One major issue with the preceding estimate of anthropogenic $CO_2$ in the ocean is that it must be separated from the natural background of DIC in the ocean. In the case of the model, this is straightforward (although there remain several ways of doing so), but it is challenging observationally. The datasets used in this study, Key et al. (2004) and Lauvset et al. (2016) use different methodologies (as well as different-sized underlying databases) to estimate and separate anthropogenic and natural $CO_2$. As

this complicates evaluation of the model's distributions, the lower row of Figure 18 shows the vertical inventory of CFC-11, a conservative artificial tracer accumulating within the ocean similarly to anthropogenic $CO_2$. Relatively straightforward to quantify to high precision, and without any natural background, this tracer serves as a loose proxy for anthropogenic $CO_2$ (Dutay et al., 2002; Doney et al., 2004). As such, it provides a second performance measure against which to compare the interior redistribution of surface anthropogenic $CO_2$ uptake. Overall, UKESM1's CFC-11 distributions better match those of

the observational dataset than anthropogenic $CO_2$. However, the same differences also arise, particularly the east-west gradient in Atlantic column inventory, likely for the same reasons suggested above. The model also exhibits more extensive coastal uptake of CFC-11 in the Weddell Sea.

### 3.5    Interior biogeochemistry

Figures 21 and 22 show intercomparisons of ocean interior DIN and DIC, with Supplementary Figures S14 to S17 showing the

corresponding intercomparisons for other tracers. Per Figure 6, the plots use a thermohaline transect to illustrate the connection between young watermasses in the North Atlantic through to old watermasses in the North Pacific.

In the case of DIN, a number of the main observational features are reproduced, including the low concentrations in the Arctic and (especially) the surface oligotrophic gyres, generally lower concentrations within the NADW, a limb of elevated concentrations within the AAIW, intermediate concentrations within the Southern Ocean, and the highest concentrations in the

North Pacific, particularly at midwater depths. However, despite this agreement on the main patterns of features, the model also exhibits a number of pronounced biases. In the Atlantic basin, near-surface positive biases overlie strong negative biases in the



upper 3 km, where maximum concentration differences of more than 10 mmol N m$^{-3}$ occur, while in deeper waters the bias is reversed to strong positive. This split of biases is generally aligned with the NADW and AABW watermasses in this basin. In the Pacific basin this pattern is broadly repeated, although with stronger positive bias in the upper 1 km, and less pronounced,
but similar sign, biases at depth. More clearly than in the Atlantic basin, the model shows a shallow focused layer of maximum DIN concentration in the upper 1 km, while observations indicate a more gradual change in DIN concentration with depth. An indication of its source lies in Supplementary Figure S15, which shows the corresponding transect for dissolved oxygen. Oxygen concentrations are typically highest at the surface where they are replenished by the atmosphere, and progressively lower in older watermasses as oxygen is consumed by remineralisation of sinking organic matter driven by the biological pump.
Based on these fields, UKESM1 exhibits a bias towards shallower remineralisation, with less nitrogen reaching the deep ocean interior through sinking particles, and corresponding overconsumption of oxygen in shallower waters, and underconsumption at depth.

Supplementary Figure S14 shows the corresponding situation for silicic acid, a nutrient which is primarily consumed by diatom phytoplankton in the ocean (and by diatom phytoplankton only in UKESM1). Unlike nitrogen, which is incorporated
in organic matter and widely used in cellular biochemistry, silicic acid is polymerised to make protective shells (frustules), and is returned to solution principally by physicochemical dissolution rather than active remineralisation (Kamatani, 1982). Consequently, its biological turnover is slower, and a greater proportion of biogenic silica (opal) reaches the deep ocean than nitrogen. Coupled to the current mode of the thermohaline circulation, which has deep-water formation in the Atlantic and the oldest watermasses in the Pacific, this results in a deep nutrient distribution where the highest concentrations occur in the
Pacific basin. UKESM1 generally reproduces the differences in the nitrogen and silicon distributions, although with a number of biases. Principally, the AABW cell in the Atlantic is a more significant reservoir of silicic acid, while the North Pacific maxima is decreased. The silicon nutricline in the North Pacific is also deeper, although this is shallower in the South Pacific. Overall, the model shows a less skewed silicon cycle, with a greater fraction of total silicon stored in the Atlantic than observed.

As Figure 22 shows, the biological coupling between nitrogen and carbon means that the distribution of DIC in the ocean
shares a number of common patterns with nitrogen (albeit against a high background concentration driven by $CO_2$ solubility). As a consequence, UKESM1's DIC distribution also shares a number of the same biases, including NADW negative biases and AABW positive biases. However, modelled DIC has an additional negative bias across the global domain, indicating that the ocean of UKESM1 has a lower mean DIC concentration than observed, 2289.4 mmol C m$^{-3}$ as compared to 2337.6 mmol C m$^{-3}$ (-2.1%; Key et al., 2004). Figure 19 shows the corresponding profiles of modelled and observed DIC,
together with corresponding profiles of alkalinity, anthropogenic $CO_2$ and CFC-11. As shown by Figures 16 and 17, this difference in DIC concentration does not prevent the model from realistically simulating the rate of ocean exchange and uptake of anthropogenic $CO_2$ over the Historical period, but it alters the model ocean's carbonate chemistry system including ocean pH, potentially with consequences (see Section 4.3).

Supplementary Figure S16 shows the corresponding distributions of alkalinity. While patterns of surface alkalinity are pri-
marily driven by the hydrological cycle (evaporation, precipitation and runoff), interior alkalinity is affected by marine biogeochemistry. In UKESM1 a simplified alkalinity cycle is represented with only the net production of calcium carbonate ($CaCO_3$;





calcite polymorph) affecting alkalinity distributions (i.e. "hard tissues pump" only, no "soft tissues pump"; cf. Marinov and Sarmiento, 2004). As this production of $CaCO_3$ is ultimately tied to the production of organic material, the patterns of bias in alkalinity overlap with those already seen. However, a significant mismatch in model alkalinity is a general negative surface

bias. As alkalinity balances dissolved $CO_2$, bicarbonate and carbonate, it regulates total DIC concentration, with a negative bias in alkalinity acting to reduce total DIC concentration. Such a negative DIC bias at the surface preconditions the interior ocean to lower DIC, consistent with Figure 22. To further illustrate this model bias, Supplementary Figure S20 shows the observed and simulated relationships between salinity and alkalinity. Each data point is a surface alkalinity versus surface salinity, and the plot shows the linear relationship between these properties (c.f. Lee et al., 2006) and the offset from the observed relation-

ship exhibited by UKESM1. The calculated regressions intersect at a salinity of 35 PSU, although model alkalinity generally lies below that observed even above this value.





## 4 Discussion

For ESMs to deliver reliable estimates of future global change, including quantification of key feedbacks, it is important that the states of their component submodels are realistic, in particular for climate-relevant time-mean distributions and temporal trends

of material (carbon) and energy (heat). Here we have examined the state of the ocean component of the UKESM1 model, a new state-of-the-art ESM, and participant in CMIP6. We have evaluated the performance of both physical and biogeochemical aspects of the ocean submodel in the context of diverse observational datasets. As well as the model's "present-day" state, we have additionally examined trends in key model properties across the Historical period (1850–2014). Kuhlbrodt et al. (subm.) presents a complementary analysis of ocean heat uptake. Seperate ensemble members have been used to understand

the consistency of these temporal trends, but where comparing with observational fields, we have used model output averaged across the Historical ensemble (per Supplementary Table S1).

In terms of the physical performance of UKESM1's ocean, its state is broadly realistic, but with a number of biases. At the ocean's surface, temperature is well-reproduced globally, but with biases including a warm Southern Ocean driven by receipt of too much shortwave radiation (Sellar et al., 2019), and a marked North Atlantic "cold spot" associated with poor

Gulf Stream separation and North Atlantic Current pathway. Model upper ocean mixing also reproduces the geographical and seasonal patterns observed, with a bias towards exaggeration of extreme low and high mixing. UKESM1's sea-ice distribution captures much of the seasonal cycle in both hemispheres, although is biased positive (and thicker) throughout the year in the north (driven primarily by excessively cooling aerosol forcing), while falling short of its maximum extent in the south (in part owing to the SO warm bias). The excess in Arctic sea-ice is driven by a general cool bias in surface temperature in the northern

hemisphere in UKESM1, a product of aerosol or land-use forcing (Sellar et al., 2019). In the ocean interior, compensating biases in temperature and salinity are found, related to the deficiencies in the overturning circulation mentioned in Section 3.1 (Figures 6 to 8), as well as a cumulative warming bias produced during forced ocean-only spin-up (Yool et al., 2020).

Biogeochemical performance of UKESM1 largely traces to previous applications of the model (e.g. Yool et al., 2013) despite a significantly longer-duration spin-up as part of UKESM1 (Yool et al., 2020). Regarding the ocean's nutrient cycles

and biological activity, the model displays a pattern of general agreement but with biases that are sometimes large. In the surface ocean, while retaining major nutrient boundaries, the model also exhibits excessive nitrogen and silicon in the Southern Ocean, excess nitrogen in the Equatorial Pacific, and depletion of silicon in the North Pacific. Upper ocean productivity in the model also follows major observed patterns, though with biases including a excessively productive Southern Ocean (both geographically and temporally), and insufficiently productive oligotrophic gyres. These biases are also mirrored in other important

biological fields such as zooplankton and in the surface concentration of dimethyl sulphide. Meanwhile, in the ocean interior, biases in mesopelagic nitrogen and oxygen indicate that remineralisation of sinking biogenic material in the model occurs too shallow, with compensating opposite-sense biases below. In the deep Atlantic, the model's sluggish AABW cell accumulates more nutrients than observed, both nitrogen and silicon, while losing more oxygen.

Regarding the ocean's carbon cycle, the model represents patterns of surface carbon properties well, although with general

negative biases in both DIC and alkalinity concentrations. Spatial and temporal patterns of air-sea $CO_2$ exchange are broadly in





agreement with those estimated from observations, though the model does not well represent Southern Ocean outgassing, and simulates excessively strong North Atlantic ingassing. Despite these discrepancies, the model falls within the uncertainty in the observationally-estimated temporal patterns of net ocean $CO_2$ uptake over the Historical period. Storage of anthropogenic $CO_2$ in the model ocean generally matches that estimated from observations, with high amounts in the Southern and North

Atlantic oceans. However, the model exhibits a spatial discrepancy in storage in the North Atlantic, with southward transport down the western side of the basin in NADW noticeably lower than observed. Within the ocean interior, because of the role of the biological pump, the spatial pattern in DIC biases tracks those of nitrogen. However, the surface bias towards lower DIC also imposes a general negative bias throughout the ocean interior, with the model ocean storing less carbon than observed in the Earth system.

On the spatial and temporal scales analysed here (i.e. global and centennial), the main fields and time-series analysed show good consistency across the UKESM1 ensemble. For higher time frequencies (e.g. decadal in the Southern Ocean; interannual for the El Niño–Southern Oscillation), or for smaller regions with significant dynamics (e.g. the Arctic), cross-ensemble variability will be more important, and will be considered for detailed future studies.

### 4.1 Biogeochemistry biases

As already described in Sections 3.1 and 3.2, UKESM1 has a number of physical biases. Examining these biases within UKESM1 forms a component of a number of parallel studies, including on circulation and Gulf Stream seperation (Kuhlbrodt et al., 2018), sea-ice thickness (SIMIP Community, 2020), ocean heat uptake (Kuhlbrodt et al., subm.), and AMOC trends (Menary et al., 2020). Consequently, in the following, we focus on explaining the biogeochemical biases found within UKESM1.

As described above, although UKESM1 reproduces the broad patterns observed in marine biogeochemistry, it also includes

a number of significant biases in properties. In the following, we consider the likely underlying causes as well as potential actions to address them in future versions of UKESM1.

Vertical profiles of nitrogen, oxygen and carbon display matching patterns driven by the action of the biological pump. Nitrogen and carbon consumed by phytoplankton growth in the upper ocean are transported as organic material by this pump into the ocean interior where they are released back to dissolved inorganic forms in parallel with the consumption of oxygen.

In UKESM1, the profile of this process is skewed, with remineralisation of organic matter occurring too shallow in the water column, resulting in excess nitrogen and carbon in the mesopelagic, a corresponding deficit of oxygen, and reversal of these biases in deeper waters that less sinking material reaches. In MEDUSA, the organic material reaching the deep interior does so primarily as "fast-sinking" particles, coupled to a ballast model in which biominerals (opal and calcite) "protect" this organic flux. Extending the remineralisation lengthscale of these sinking particles, or affording them greater biomineral protection, are

both means of addressing this bias to first order.

Significantly for ocean productivity, UKESM1's ocean displays strong positive biases in the surface concentration of nitrogen nutrient in a number of ocean regions, including the Southern Ocean, the Equatorial Pacific and the Peruvian Upwelling. Such biases can indicate oversupply of nutrients, or insufficient consumption by phytoplankton. On the first, as all three regions experience significant upwelling of interior waters, any biases in the nitrogen supply from these watermasses will play a role.





For instance, shallow remineralisation bias noted above will contribute toward the positive biases in surface waters in these regions. On the second, an additional issue lies with the availability of the micronutrient iron in these regions. Although this is also supplied by upwelling watermasses, its availability is also dependent on deposition of iron from aeolian dust, and this deposition is biased negative in these regions in UKESM1. While atmosphere-land aspects of the deposition flux may ultimately be important here (e.g. location of desert source regions, patterns of wind dispersal), within the ocean model itself, parameter

changes to reduce iron stress (more iron from dust, lower iron quotients in phytoplankton) could assist here. However, by relieving iron stress in this uniform way, there may be consequences elsewhere in the model.

Leaving aside the interior biases described above, carbon in UKESM1 is more generally negatively biased throughout the ocean, with implications for the ocean's role as the largest reservoir of carbon in the Earth system. As noted previously, surface alkalinity plays a role in interior carbon by buffering the surface carbonate system and regulating the surface DIC concentrations

that ultimately ventilate the ocean interior. Modelled surface alkalinity has a general negative bias, and a different relationship with surface salinity than that observed (cf. Lee et al., 2006). In the model, aside from hydrological cycle processes, only net calcium carbonate production (and its subsequent dissolution at depth) affects alkalinity, and this acts to decrease its upper concentration and increase its interior concentration below the calcite compensation depth (CCD). As such, this bias could be addressed in MEDUSA simply by decreasing calcium carbonate production (and its export) to increase the retention of

alkalinity in the surface ocean. However, while alkalinity is generally lower across the upper ocean, calcite production is not uniform, with a latitudinal gradient in which most net calcification occurs in the tropics. A broader point is that calcium carbonate production in MEDUSA is highly simplified and only concerns the fraction export to the ocean interior, whereas other models treat it in more complex ways (e.g. Kvale et al., 2015; Butenschön et al., 2016; Buitenhuis et al., 2019) that potentially offer more realistic solutions than simple parameter scaling.

Another clear surface bias in UKESM1, and one which is easy to discern because of the ready availability of synoptic, high quality observational data, is its field of surface chlorophyll. In the Southern Ocean in particular, the seasonal spring-summer bloom has higher chlorophyll concentrations that persist longer and extend further polewards. Even in winter, anomalously high chlorophyll concentrations ($> 0.1$ mg m$^{-3}$) extend southward to the tip of the Antarctic Peninsula. This bias is strongly associated with a corresponding productivity bias, although at the highest latitudes (in both hemispheres) there is a degree of

decoupling. This bias is particularly significant in UKESM1 because simulated chlorophyll is used in its empirical submodels of DMS and PMOA, both of which are climatically-active compounds (cf. Quinn and Bates, 2011). Noticeably, the high concentrations of chlorophyll simulated at high latitudes also persist beyond the peak of productivity. In part these biases are related to negative sea-ice biases that allow more light to penetrate into the high latitude ocean, but their excess extent and persistence also suggest that the chlorophyll submodel may be too responsive under low light conditions. At lower latitudes,

where light is less limiting and nutrient stress more important, sensitivity to the chlorophyll model is less pronounced.

Separate from these biogeochemical biases, the model exhibits several physical biases, including a general warm bias throughout the ocean, warm and cool biases regionally, some hemisphere-specific ice biases, and issues with interior circulation. These all affect the realism of the physical regime in which MEDUSA's biogeochemistry is embedded and introduce biases independently of those arising from its deficiencies. For instance, the weaker deep overturning cell north of the ACC





combined with a NADW cell which is slightly too weak (Figure 8) leads to a colder and fresher deep ocean. At the same time, these circulation features favour the build-up of nutrients and the corresponding depletion of oxygen.

Finally, the corrective measures outlined above are proposed independently without any consideration of their full impacts. For instance, decreased calcium carbonate production is proposed as a countermeasure to decrease the negative bias in surface alkalinity. However, this change will also decrease the quantity of sinking organic material "protected" by this mineral, allowing
it to be remineralised more rapidly, shoaling the remineralisation horizon of the biological pump, and worsening the biases in nitrogen, carbon and oxygen profiles. This interdependency of model biogeochemical processes and states – and their dependency on the ocean physical state – significantly complicates model tuning, particularly given the long timescales of ventilation and three-dimensional connectivity in the ocean. Optimisation techniques such as the Transport Matrix Method (TMM; Khatiwala et al., 2005) are increasingly being used to address this (e.g. Kriest, 2017), although the resulting solutions
are also found to be sensitive to the physical framework (Kriest et al., accepted).

### 4.2   CMIP intercomparison

Figure 23 and Supplementary Figures S21 to S26 illustrate the performance of UKESM1 alongside a series of CMIP6 models for the same suite of key surface biogeochemical properties already shown. Annual mean fields for each property for each model are shown, together with the corresponding observational field (data missing from the CMIP6 archive is denoted by a blank
field). Fields are also shown from UKESM1's CMIP5 predecessor, HadGEM2-ES (Totterdell, 2019), to illustrate improvement between CMIP generations, together with those from MEDUSA-2.0 (Yool et al., 2013) to demonstrate the traceability of UKESM1's MEDUSA-2.1 to prior work (note that this latter work is ocean-only rather than fully-coupled).

The CMIP6 models included in this analysis are: CESM2-FV2 (Danabasoglu et al., 2020); CNRM-ESM2-1 (Voldoire et al., 2019); CanESM5 (Swart et al., 2019); IPSL-CM6A-LR (Boucher et al., 2020); MIROC-ES2L (Hajima et al., 2020); MPI-
ESM1-2-LR (Mauritsen et al., 2020); MRI-ESM2-0 (Yukimoto et al., 2019); NorESM2-LM (Tjiputra et al., 2020).

Supplementary Figures S21 and S22 show patterns of surface nitrogen and silicon nutrients. Reassuringly, most of the models capture the main geographical features of availability, including high abundance in the Southern Ocean and the subpolar north, low availability throughout the subtropics, and elevated concentrations in upwelling regions (less prominently in the case of silicon). All of the models do display biases, however, differing in over- or under-estimation of Southern Ocean concentrations
(UKESM1 consistently over-estimates), and in how low subtropical concentrations are drawn down to.

The patterns in surface chlorophyll shown in Supplementary Figure S23, however, are more diverse. As already noted, UKESM1 exhibits both excess concentrations in regions such as the Southern Ocean and Equatorial Pacific, and negative biases in its oligotrophic gyre regions. Other CMIP6 models exhibit both similar and different biases. For instance, several models share UKESM1's positive biases in major productive regions (MPI, MRI), while others reverse its pattern in oligotrophic region
and instead have excessive chlorophyll concentrations (MIROC, NorESM). In general, while patterns of surface nutrient are broadly shared by models, chlorophyll patterns are instead somewhat divergent.

Figure 23 shows a similarly diverse pattern for ocean productivity, with models estimating both much higher and much lower global totals. While all of the models show biases, they agree on the focusing of productivity in key biomes such as the





temperate high latitudes and upwelling regimes, although the biases found are not always aligned with those in chlorophyll.
Excessive productivity in the Southern Ocean is significant problem in UKESM1, although it is noticeable that models using PISCES marine biogeochemistry (IPSL, CNRM) do much better in this regard.

Supplementary Figures S24 and S25 respectively show surface DIC and alkalinity across the suite of models. As already suggested from the results of UKESM1, biases in surface alkalinity are important in setting biases in DIC, with several models showing matching positive biases in both (CanESM, MRI). Interestingly, while UKESM1's institutional precursor model,
HadGEM2-ES, shares neither its ocean physics nor its marine biology (Totterdell, 2019), the models share biases, particularly in alkalinity, a field strongly governed by atmospheric freshwater interactions, and a component where the models do share submodels. This underscores the role that other Earth system components may play in shaping model marine biogeochemistry.

Similarly, there is generally strong agreement in patterns of air-sea $CO_2$ flux shown in Supplementary Figure S26. The models broadly reproduce the latitudinal patterns of flux observed, outgassing in the tropics and (generally) ingassing at high
latitudes. The models differ in detail, with variation in the magnitude of $CO_2$ uptake in regions such as the North Atlantic, its release along the Equatorial Pacific, and in the magnitude and geographical extent of outgassing regions in the Southern Ocean. Interestingly, a marked bias in UKESM1, strong outgassing along the west coast of South America, is reproduced in several models (CanESM, MRI), while being absent in others (MIROC, CESM).

Figure 24 summarises the performances of this suite of models using Taylor diagrams Taylor (2001). In each case, the panels
indicate spatial variability normalised to that of observations (radial axis) and model-observation correlation (circular axis), both at the global, annual mean scale used in the preceding figures. In such diagrams, proximity to the red and black circle on the x-axis indicates agreement with the observational field. Overall, UKESM1 performs comparably with other ESMs, particularly well for DIC and alkalinity, and less well for DIN. The panels also show that no one model is superior in all properties, with the "best" model differing between properties, and that the various models tend to perform similarly across
properties. Chlorophyll, in particular, is a property that all of the models perform badly at, while DIN is something they all perform relatively well at.

Note that this cross-CMIP6 analysis overlooks the role played by the duration of spin-up prior to Historical simulations in the magnitude of model biases. The analysis Séférian et al. (2016) found that spin-up duration of CMIP5 models ranged widely from 200 years up to almost 12000 years, and that this duration could explain the magnitude of biases. Essentially, the
longer that a model is spun-up, the greater its drift from the observationally-derived initial conditions that also typically serve as performance targets (as they do here). In the specific case of UKESM1, its ocean component was spun-up for approximately 5300 years to equilibrate its net air-sea $CO_2$ flux below a target of 0.1 Pg C y$^{-1}$ (Yool et al., 2020). This duration was also sufficient for other physical and biogeochemical properties to approach quasi-equilibrium, and UKESM1's performance is unlikely to be significantly affected by drift.

In terms of performance between CMIP generations, UKESM1 shows improved representation across almost all properties relative to HadGEM2-ES, with the exception of surface DIN (where excess concentrations in the Equatorial Pacific impact UKESM1's global realism). UKESM1 improves on the marked biases in silicic acid and chlorophyll in particular, as well as a generally better representation of the ocean's role in $CO_2$ exchange. Séférian et al. (2020) provides a more complete view of



the improvements achieved in marine biogeochemistry modelling from CMIP5 to CMIP6, including between HadGEM2-ES
and UKESM1.

Finally, the preceding figures show good traceability in the marine biogeochemistry performance of UKESM1 with previous
instances of its use (e.g. Yool et al., 2013). For better and for worse, UKESM1 and MEDUSA-2 perform similarly across all of
the properties examined. With the exception of DIN, where UKESM1's geographical biases are similar but clearly larger than
those of MEDUSA-2, UKESM1's performance in Figure 24 is marginally better (and despite a much longer spin-up period;
5300 vs. 120 years).

## 4.3 Future projection

As described above, when compared to observational metrics, UKESM1 performs well over a large number of diverse physical
and biogeochemical properties. However, the model displays a number of biases in the present-day state that have implications
for its future behaviour under different climate scenarios.

UKESM1's Arctic sea-ice is biased positive in both seasonal extent and, in particular, thickness. In the absence of biases in
the other direction, these aspects will enable it to persist longer under climate change, with a range of likely consequences for
the Arctic environment (Thackeray and Hall, 2019).

Decreased productivity is a common ecosystem response under climate change, as ocean warming enhances ocean strati-
fication, reduces nutrient resupply from mixing and depletes surface concentrations (Kwiatkowski et al., 2020). The positive
nitrogen nutrient biases across UKESM1's ocean may (at least temporarilty) stave off this depletion dampening the response
of its marine ecosystem. In particular, the excess nutrient bias in the Arctic may result in unrealistic future responses as the
Arctic continues to thaw (cf. Popova et al., 2012; Vancoppenolle et al., 2013).

In terms of surface DIC and alkalinity, UKESM1 performs best in the CMIP6 ensemble examined here (Figure 24). However,
as already noted, UKESM1 exhibits a negative bias in surface alkalinity which drives a corresponding bias in surface DIC (and
within the ocean interior more generally; Figure 22). This bias reduces the buffering capacity of the surface ocean (Egleston
et al., 2010) and impacts the long-term capacity of the model ocean to act as a reservoir for carbon (cf. Archer, 2005).

Staying with the carbon cycle, although the simulated uptake of anthropogenic $CO_2$ by the ocean is comparable to that
estimated at the global scale (Figure 17, its spatial pattern within the ocean interior exhibits circulation-driven biases (Figure
19). Invasion of anthropogenic $CO_2$ into shallow or rapidly ventilated water masses will lead to its more rapid return to the
surface ocean and atmosphere, potentially reducing future uptake by the ocean.





# 5 Conclusions

– Physical and biogeochemical properties of the ocean component of the UKESM1 model have been evaluated against observations for the Historical period.

– Examined properties indicate that the model generally reproduces the main geographical and temporal features of the ocean, but with a number of marked biases.

– Physically, model biases include a global warm bias, resolution-dependent surface biases, excessive northern sea-ice, sluggish AABW circulation and weak DWBC flow.

– Biogeochemically, model biases include nutrients skewed by remineralisation and iron availability, corresponding productivity biases, and surface chemistry causing reduced carbon storage.

– Temporally, the ocean shows a number of secular trends including aerosol-driven strengthening Atlantic MOC transport, associated sea-ice and productivity changes, and realistic carbon uptake.

– The UKESM1 ensemble shows consistent behaviour across ocean properties, it performs well in key metrics compared to CMIP6 peers, and improves on that of its CMIP5 predecessor, HadGEM2-ES.

– Though overall performance is good, UKESM1's biases have implications for its response to climate change, including the sea-ice loss rate, future productivity changes and ocean carbon uptake.





*Code and data availability.* All simulations used in this work were performed using version 10.9 of the Unified Model (UM), version 5.0 of JULES, NEMO version 3.6, CICE version 5.1.2 and OASIS3-MCT version 3.0. Model output from the NEMO ocean model was handled using the XML Input-Output Server (XIOS) library (Meurdesoif, 2013).

Guidance concerning the availability and use of UKESM1 is available from a dedicated website:

`http://cms.ncas.ac.uk/wiki/UM/Configurations/UKESM`

(last access: 30 October 2020)

Due to intellectual property rights restrictions, neither the source code nor documentation papers for the UM or JULES can be provided. However, the Met Office UM is available for use under licence, and further information on how to apply for a licence is available here:

`https://www.metoffice.gov.uk/research/approach/modelling-systems/unified-model/`

(last access: 30 October 2020)

JULES is also available under licence, free of charge, with further information on obtaining access for research purposes here:

`http://jules-lsm.github.io/access_req/JULES_access.html`

(last access: 30 October 2020)

The simulation data used in this study are archived on the Earth Sytem Grid Federation (ESGF) node:

`https://esgf-index1.ceda.ac.uk/projects/cmip6-ceda/`

(last access: 30 October 2020)

The model Source ID for UKESM1 is UKESM1-0-LL, and simulations are identified by the following Variant Labels: r5i1p1f2, r6i1p1f2, r7i1p1f2, r4i1p1f2, r8i1p1f2, r1i1p1f2, r2i1p1f2, r3i1p1f2, and r9i1p1f2 (see Supplementary Table S1 for more details). The simulation data are also archived at the Met Office and are available for research purposes through the JASMIN platform (`www.jasmin.ac.uk`). For further

details please contact `UM_collaboration@metoffice.gov.uk` referencing this paper.

The Matlab scripts used for analysis and plotting are available in this Zenodo archive:

`https://doi.org/10.5281/zenodo.4155210`

(last access: 30 October 2020)



## Appendix A: MEDUSA-2.1

MEDUSA's dual size-structure resolves small (nanophytoplankton and microzooplankton) and large (microphytoplankton and mesozooplankton) components. Similar to its living components, MEDUSA's detrital components are also split into two size classes, with small, slow-sinking detrital particles represented explicitly as separate nitrogen and carbon tracers, and large, fast-sinking particles represented implicitly. At the seafloor, MEDUSA resolves 4 reservoirs to temporarily store organic (nitrogen and carbon) and inorganic (opal and $CaCO_3$) material reaching the sediment via both slow- and fast-sinking particles (iron is

slaved to nitrogen in these reservoirs). Supplementary Figure S2 presents a schematic outline of MEDUSA's components and the process connections between them.

The model's nitrogen, silicon and alkalinity cycles are closed and conservative (e.g. no riverine inputs), while the other three cycles are open. The ocean's iron cycle includes additions from aeolian and benthic sources, and is depleted by scavenging based on local iron availability (and an assumed fixed binding ligand concentration). The ocean's carbon cycle exchanges $CO_2$

with the atmosphere based on local carbonate chemistry, atmospheric $xCO_2$ and ambient winds. The ocean's oxygen cycle exchanges with the atmosphere (which has an assumed fixed oxygen concentration), and dissolved oxygen is additionally created by primary production and depleted by remineralisation throughout the ocean. The various elemental cycles include both fixed and variable stoichiometry. Iron is slaved to nitrogen throughout, while nitrogen and carbon have fixed (but different) ratios in phytoplankton and zooplankton, and variable ratios in detritus. Diatom silicon has a variable ratio with nitrogen,

dependent on nutrient availability and growth rate. Calcium carbonate is produced at a geographically-variable rate relative to organic carbon according to the ambient calcite saturation state, and consumes both dissolved inorganic carbon (DIC) and alkalinity in a ratio of 1:2 respectively. Oxygen production and consumption reflects the C:N ratio of organic matter produced and consumed.

Yool et al. (2013) extensively describes the structure, differential equations, functional forms and parameterisation of the

MEDUSA-2.0 model in an earlier, ocean-only configuration. As part of the development cycle of UKESM1, a number of changes were made to the model, and the resulting version used here is denoted as MEDUSA-2.1 for clarity. These specific developments are listed below.

- The carbonate chemistry submodel used in MEDUSA-2, Blackford et al. (2007) (also Artoli et al., 2012) has been replaced by the MOCSY-2.0 scheme of Orr and Epitalon (2015). See Appendix B for more details.

- Since UKESM1 represents atmospheric chemistry, including elements of the sulphur cycle, MEDUSA now includes several empirical submodels of surface dimethyl sulphide (DMS) concentration to permit this Earth system feedback. See Appendix C for more details.

- During development and testing, a small number of changes have been made to MEDUSA parameter values.

- MEDUSA's underlying model code has been extensively reorganised into small subroutines with discrete functionality,
to facilitate better code management and to adopt newer Fortran conventions. Its code has also been reorganised to reflect changes within the host NEMO code, for example around model restarting.





– Throughout MEDUSA, processes involving the model's representation of vertical space, including the explicit sinking of slow detritus and the time-stepping of material fluxes into and out of the benthic reservoirs, have been revised to reflect the adoption of variable volume (VVL) by the host NEMO model.

– Diagnostic output in MEDUSA has been upgraded to utilise the XML Input-Output Server (XIOS) adopted by NEMO. Available output from MEDUSA has been extended to include additional diagnostics, including those requested by CMIP6.

**Appendix B: MOCSY-2**

As indicated above, MEDUSA-2.1 replaces an existing carbonate chemistry submodel with that of MOCSY-2.0 Orr and Epi-
talon (2015). This includes an improved iterative solver, applicability over a wider range of ambient conditions, as well as revised parameterisations that avoid several approximations in the earlier scheme. MOCSY-2.0 is primarily used to calculate surface ocean carbonate chemistry and air-sea $CO_2$ exchange, but it is additionally used on a periodic basis (monthly) to calculate ocean interior carbon chemistry. In MEDUSA-2.1, the latter is used to determine the dissolution depth of sinking biogenic calcite (via its normalised saturation state, $\Omega_{calcite}$). MEDUSA-2.1 principally passes bulk ocean temperature, salinity and
concentrations of dissolved inorganic carbon and alkalinity, together with atmospheric pressure, gas transfer velocity (calculated from wind speed), and $xCO_2$ (i.e. mole fraction; ppm) to MOCSY-2.0. Ocean concentrations of ambient silicic acid and estimated phosphate (= DIN ÷ 16) are additionally passed to MOCSY-2.0 for use in secondary coefficients, and for interior carbonate chemistry, depth and latitude are used to calculate pressure. MOCSY-2.0 has been implemented within MEDUSA in a "plug–and–play" manner to permit easy replacement with future revisions.

Per the guidance of Orr et al. (2017), the updated gas exchange scheme of Wanninkhof (2014) is used to calculate gas transfer velocity.

**Appendix C: DMS concentration**

As already mentioned, one addition to MEDUSA-2.1 is a representation of surface dimethyl sulphide concentration. This concentration is passed to the atmospheric chemistry component, UKCA, where it is used in UKESM1's sulphur cycle.
MEDUSA-2.1 includes four empirical calculations for surface DMS: Anderson et al. (2001), Simo and Dachs (2002), Aranami and Tsunogai (2004) and Halloran et al. (2010). After evaluation (Sellar et al., 2019), the formulation of Anderson et al. (2001) selected for use in UKESM1 simulations. This calculates DMS from three fields provided by MEDUSA-2: surface chlorophyll, $C$ (mg chl m$^{-3}$), surface daily average shortwave radiation, $J$ (W m$^{-2}$), and surface nutrient limitation, $Q$ (–). Surface chlorophyll, $C$, is the sum of contributions of the two phytoplankton types, while $J$ is provided by UKESM1's
atmospheric component. The $Q$ term is a conventional hyperbolic function of nitrogen nutrient concentration and uptake half-saturation concentration, here using the lower half-saturation concentration of the non-diatom phytoplankton, $k_{N,Pn}$ (which





has the same numerical value as that originally used in Anderson et al., 2001). Anderson et al. (2001) used these terms in a "broken stick" regression:

if $\log_{10} (C \cdot J \cdot Q) \leq s$

785        DMS = a

else

        DMS = $b \cdot [\, \log_{10} (C \cdot J \cdot Q) \text{-} s \,] + a$

Parameters $a$, $b$ and $s$ were originally fitted using an observational dataset (Kettle et al., 1999), and were tuned during the development of UKESM1 to balance the top-of-atmosphere (TOA) radiation (Sellar et al., 2019). Parameter $a$ was lowered to
1.0 (from 2.29), in line with Anderson et al. (2001)'s own assessment of likely high-biased observations; parameter $b$ was left unchanged (at 8.24); while parameter $s$ was linearly extended to 1.56 (from 1.72) to align with the reduced $a$.

## Appendix D: Observational data sources

The following weblinks are to sources of the observational data used in the evaluation of UKESM1.

– World Ocean Atlas 2013: temperature, salinity, nutrients, oxygen

795        `https://www.nodc.noaa.gov/OC5/woa13/`

– Hadley Centre Sea Ice and Sea Surface Temperature (HadISST.2.2): SST, sea-ice

        `https://www.metoffice.gov.uk/hadobs/hadisst2/`

– National Sea Ice Data Centre: sea-ice thickness and index

        `https://nsidc.org/data/G10006/versions/1`

800        `https://nsidc.org/data/G02135/versions/3`

– Estimating the Circulation and Climate of the Ocean (ECCO) V4r4: ocean circulation

        `https://ecco-group.org/products-ECCO-V4r4.htm`

– RAPID-MOCHA array: AMOC strength

        `https://www.rapid.ac.uk/data.php`

– Oregon State University Ocean Productivity group: chlorophyll and productivity

        `http://orca.science.oregonstate.edu/1080.by.2160.monthly.hdf.chl.seawifs.php`

        `http://orca.science.oregonstate.edu/1080.by.2160.monthly.hdf.vgpm.m.chl.m.sst.php`

        `http://orca.science.oregonstate.edu/1080.by.2160.monthly.hdf.eppley.s.chl.a.sst.php`

        `http://orca.science.oregonstate.edu/1080.by.2160.monthly.hdf.cbpm2.s.php`

– Rödenbeck et al. (2013): ocean $pCO_2$ and air-sea $CO_2$ flux

        `https://www.bgc-jena.mpg.de/CarboScope/?ID=oc`





- Lana et al. (2011): surface DMS

  `https://www.bodc.ac.uk/solas_integration/implementation_products/group1/dms/documents/dms-1degrex1degree.`
  `zip`

– Global Ocean Data Analysis Project: carbon chemistry and CFC-11

  `https://www.ncei.noaa.gov/access/ocean-carbon-data-system/oceans/glodap/`

  `https://www.glodap.info/index.php/mapped-data-product/`

- Moriarty and O'Brien (2013): zooplankton biomass

  `http://dx.doi.org/10.1594/PANGAEA.777398`



*Author contributions.* AY led the analysis and wrote the manuscript. CGJ, LdeM, TK, EEP, AJGN, JH, and EWB provided significant text, analysis and/or figures for the manuscript. JP, AAS, ATB, and ACC made significant contributions to the development and implementation of marine biogeochemistry and physics within the ocean component of UKESM1. All co-authors contributed invaluable comments during the drafting of the manuscript.

*Competing interests.* The authors declare that they have no conflict of interest.

*Acknowledgements.* The authors are extremely grateful for the input and support provided by colleagues within the UKESM core team at the Met Office and associated NERC research centres. The development of UKESM1, as well as its simulation for CMIP6, has involved an extended period of close collaboration by this team over several years, and the work presented here would not have been possible without this support.

AY, JP, CGJ, LdM, EEP, and TK were supported by the National Environmental Research Council (NERC) National Capability Science
Multi-Centre (NCSMC) funding for the UK Earth System Modelling project: CGJ and TK through grant NE/N017978/1; AY, JP, LdM and EEP through grant NE/N018036/1. AS was supported by the Met Office Hadley Centre Climate Programme funded by BEIS and Defra. ATB, AGN, ACC and JH were supported by NERC project CLASS (NE/R015953/1). CGJ, TK, AY, JP and EEP additionally acknowledge the EU Horizon 2020 CRESCENDO project, grant number 641816. AY further acknowledges support from the ARISE (NE/P006000/1) and Arctic PRIZE (NE/P006078/1) projects as part of the Changing Arctic Ocean program, funded by NERC.
The authors are grateful for the use of the Monsoon2 system, a collaborative facility supplied under the Joint Weather and Climate Research Programme (JWCRP), a strategic partnership between the Met Office and the Natural Environment Research Council.

The authors are additionally grateful to S. Khatiwala (air-sea $CO_2$ flux time-series estimates), S. Lauvset (GLODAPv2), R. Lumpkin (overturning streamfunction), and T. Tyrrell (carbonate chemistry) for their assistance with data products and analysis.





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



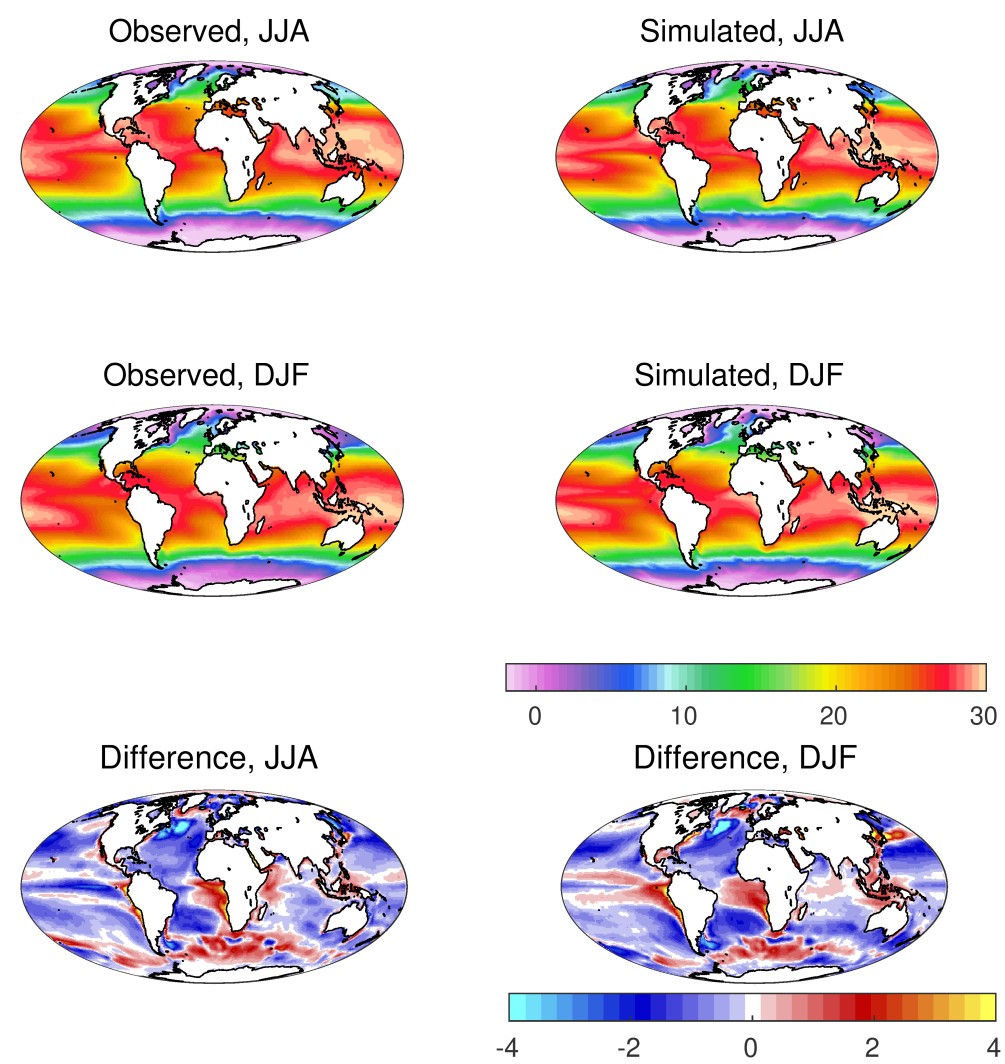

**Figure 1.** Observational (HadISST) and simulated sea surface temperature for northern (top; JJA) and southern (medium; DJF) summer.
Differences (simulated - observed) for both seasons shown in bottom row. Temperature (and difference in temperature) in °C.





**Figure 2.** Observational (left; HadISST) and simulated (right) maximum annual sea-ice cover for the Arctic (March) and Antarctic (September). Sea-ice cover is non-dimensional, and values less than 0.15 have been masked. The bottom row shows the seasonal sea-ice extent ($>$ 15% cover; in $10^6$ km$^2$) for the polar regions of each hemisphere.

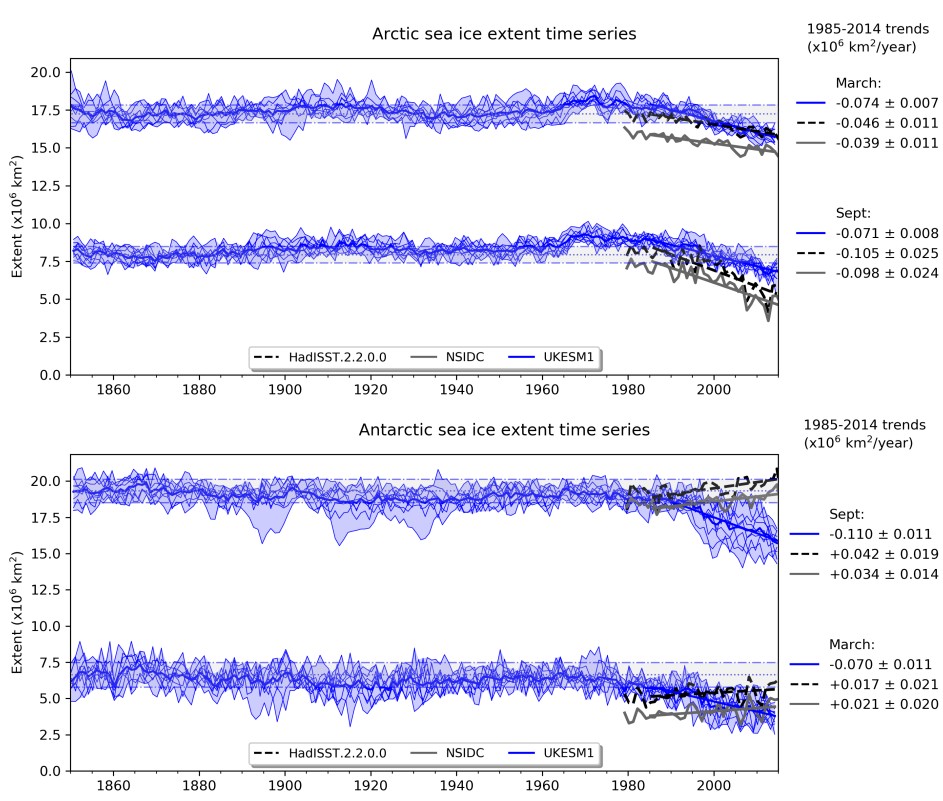

**Figure 3.** Observational (black, HadISST; grey, NSIDC) and simulated (blue) sea-ice extent in the Arctic (top) and Antarctic (bottom) across the Historical period (1850–2014). Panels show extent at seasonal minima and maxima, with recent (1985–2014) trends shown. All model ensemble members are shown, with the span of their variability shaded in blue.





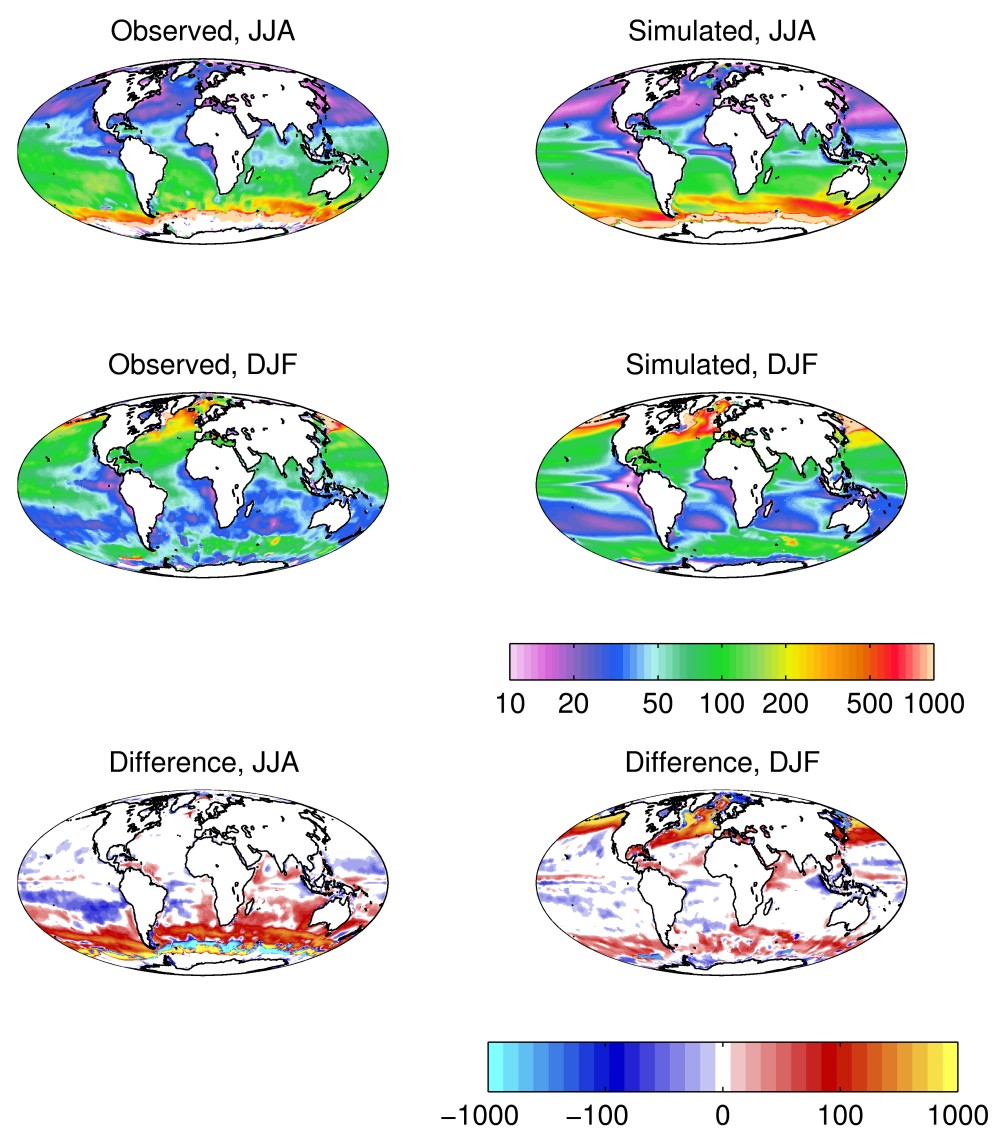

**Figure 4.** Observationally-derived (left; World Ocean Atlas) and simulated (right) mixed layer depth for northern summer (top; JJA) and southern summer (bottom; DJF). Mixed layer depth derived using a 5 m temperature criterion (0.5 °C) and full three-dimensional fields of potential temperature (Monterey and Levitus, 1997). White regions are those where this criterion fails (i.e. temperature below 5 m is never cooler by 0.5°C). Mixed layer depth in m, and shown on a logarithmic scale.





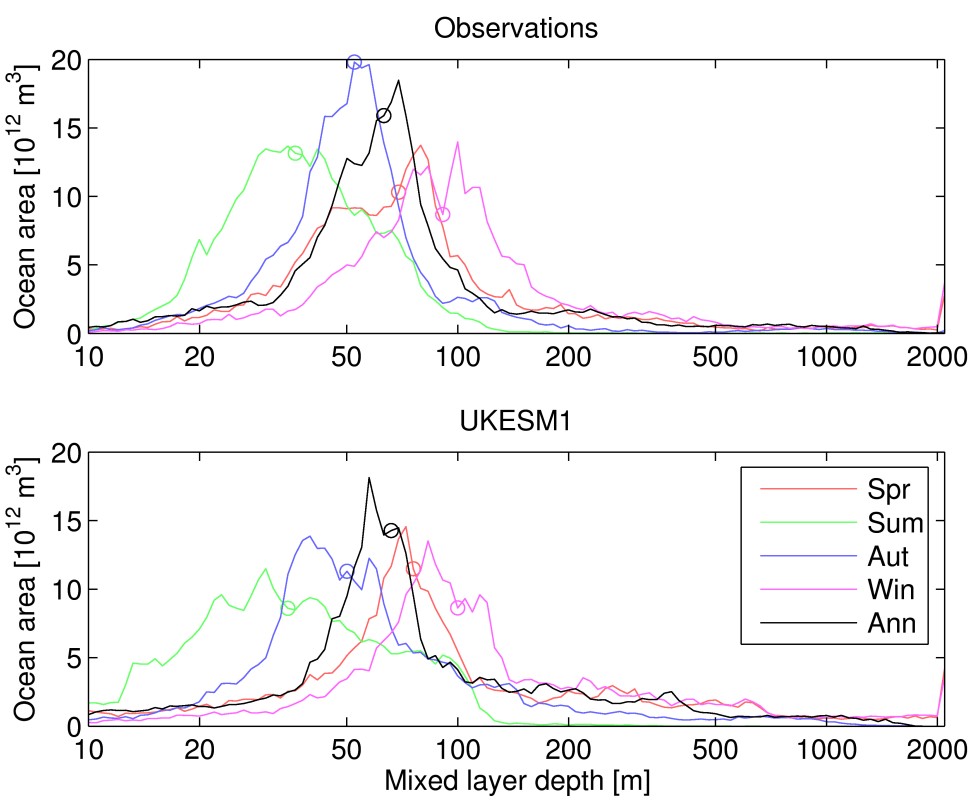

**Figure 5.** Frequency (in areal terms) of observation-derived (top; WOA) and simulated (bottom) seasonal mixed layer depths. Mixed layer depth derived here using a 5 m temperature criterion (0.5 °C) and full three-dimensional fields of potential temperature (Monterey and Levitus, 1997). Hemispheres have been temporally-aligned so that seasons co-occur (i.e. summer is JJA for the north and DJF for the south). Circles indicate the medians for each seasonal period (i.e. the 50% of ocean area mark).



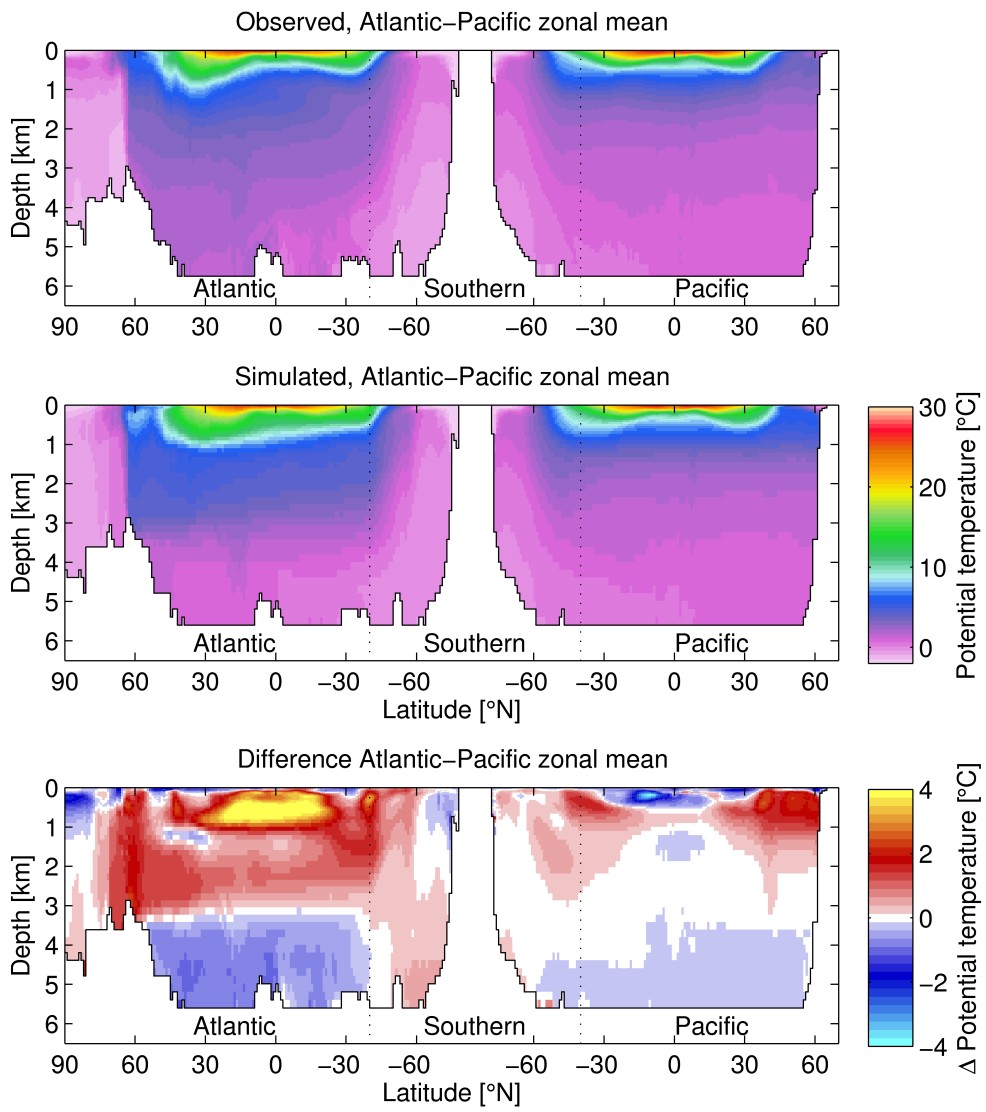

**Figure 6.** A "thermohaline circulation" section of observed (top) and modelled (bottom) zonal average potential temperature. The section tracks southwards "down" the Atlantic basin from the Arctic to the Southern Ocean, before tracking northwards "up" the Pacific basin from the Southern Ocean to the Bering Straits. The aim is to capture the stereotypical transport of deep water from its formation as a "young" water mass in the high North Atlantic through to end as an "old" water mass in the North Pacific. Dotted lines mark the "boundaries" of the Southern Ocean at 40°S in each basin. Potential temperature in °C.

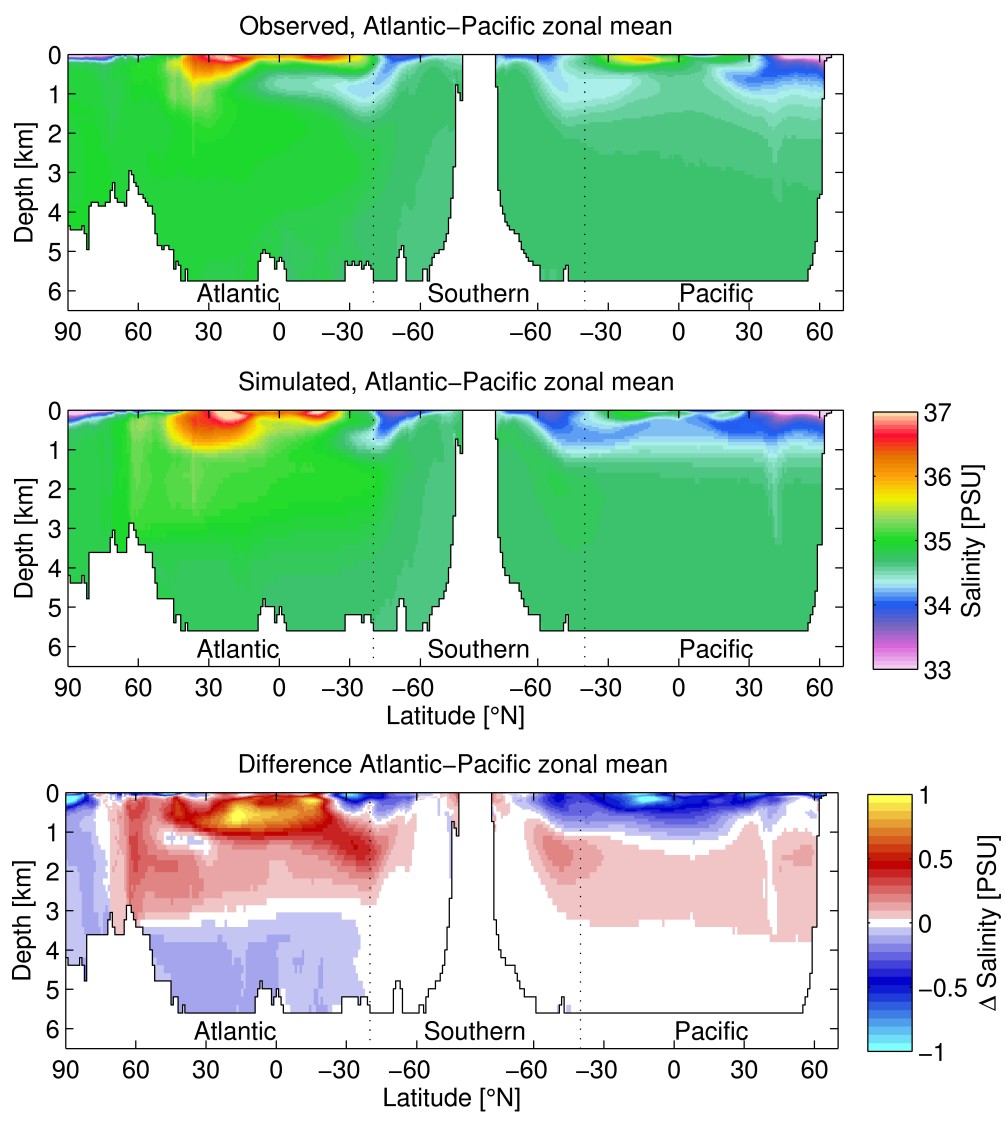

**Figure 7.** A "thermohaline circulation" section of observed (top) and modelled (bottom) zonal average salinity. Figure 6 explains the format of this section. Salinity in practical salinity units (PSU).



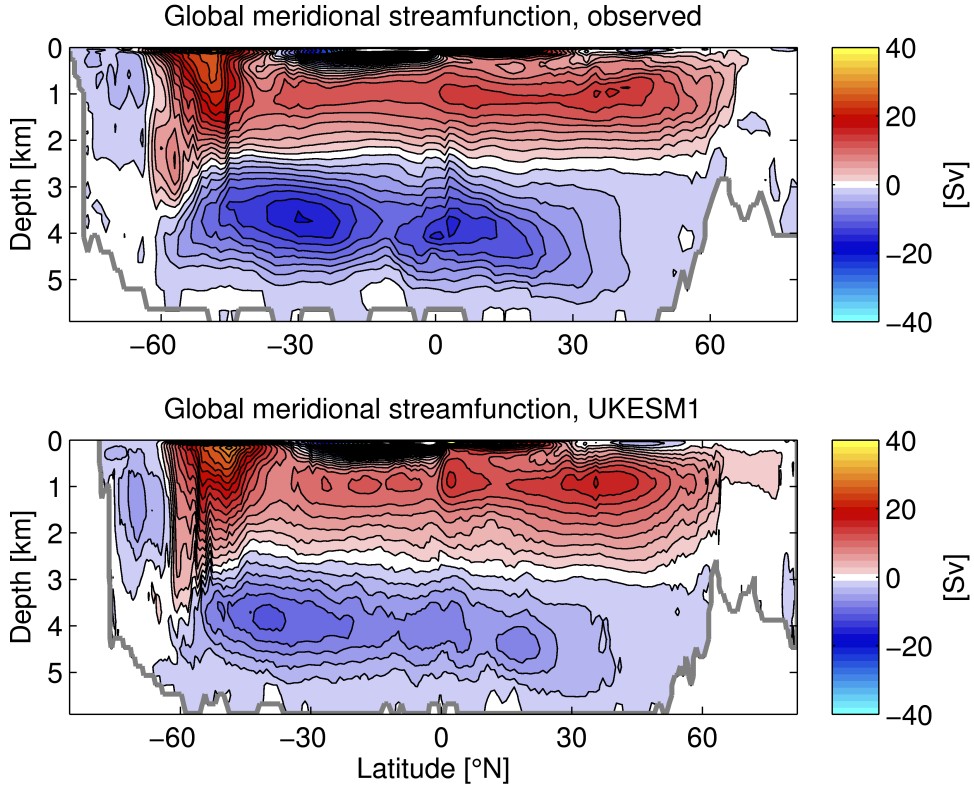

**Figure 8.** Observationally-derived (top) and simulated (bottom) meridional overturning circulation (MOC) for the global ocean. The model circulation shown is based on the decadally-averaged streamfunction. MOC in Sv with both plots including the components from parameterised mesoscaled eddies (Gent and McWilliams, 1990; Gent et al., 1995). Observational data from the ECCO V4r4 ocean circulation reanalysis for the period 1992-2017.



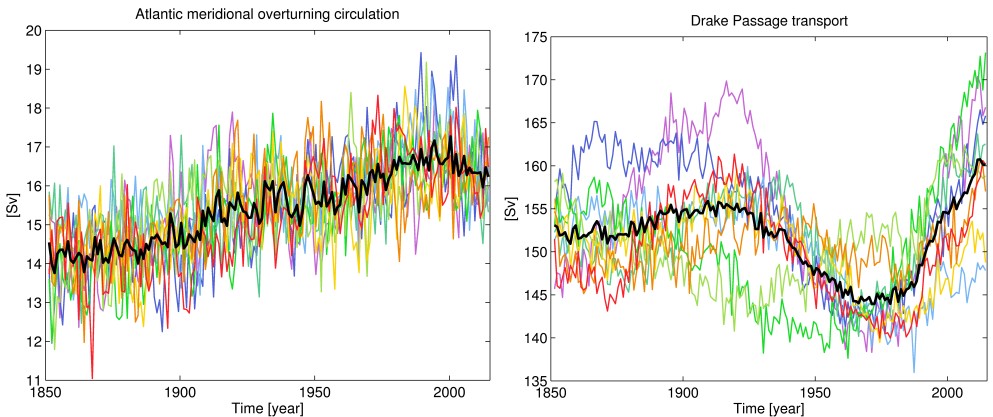

**Figure 9.** Time-series plots of the ocean circulation during the Historical period from 1850 to 2015. Panels show annual averages of AMOC (left) and Drake Passage (right) transport for all 9 ensemble members (coloured lines) and the ensemble mean (solid black line).





**Figure 10.** Observational (left; World Ocean Atlas) and simulated (right) surface dissolved inorganic nitrogen, shown geographically for northern (top; JJA)and southern summer (middle; DJF), and as zonal Hovmöller diagrams (bottom). Concentrations in mmol N m$^{-3}$.





**Figure 11.** Observational (left; World Ocean Atlas) and simulated (right) surface dissolved silicic acid, shown geographically for northern (top; JJA)and southern summer (middle; DJF), and as zonal Hovmöller diagrams (bottom). Concentrations in mmol Si m$^{-3}$.

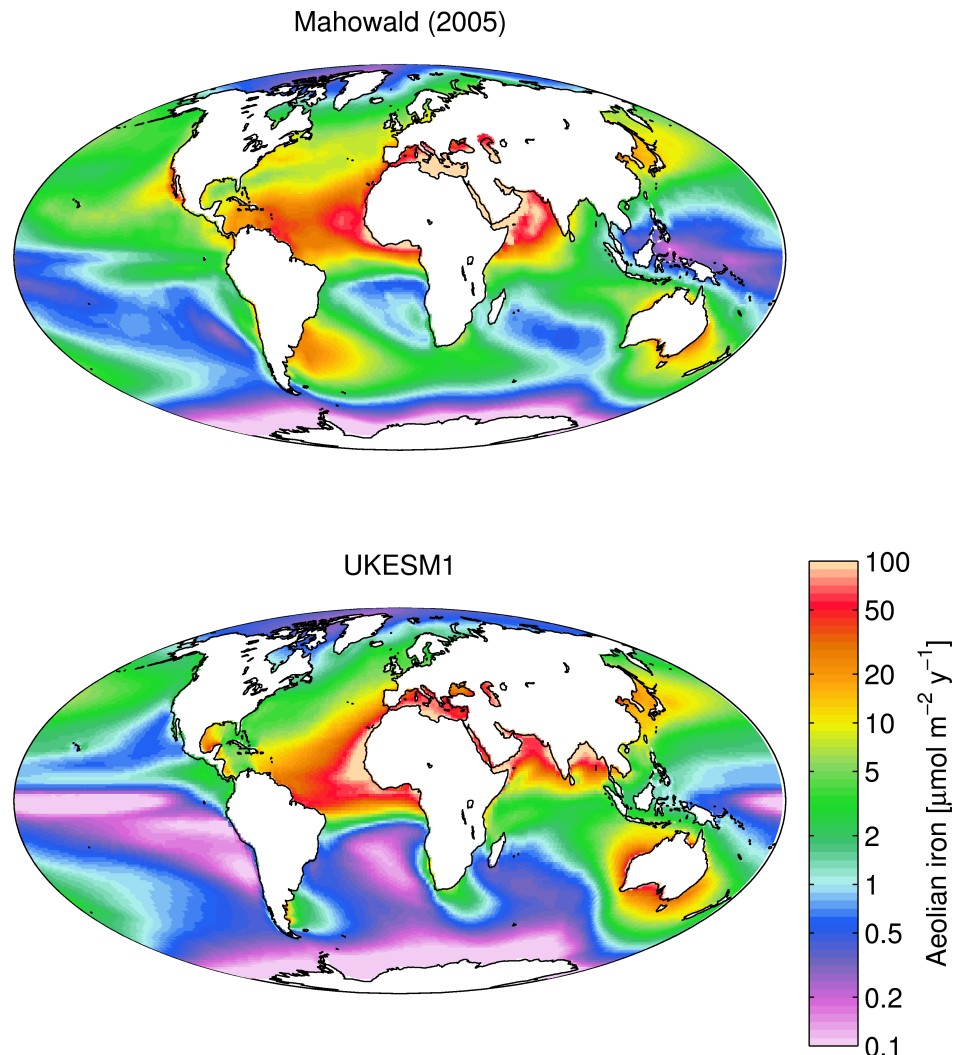

**Figure 12.** Observational (top; Mahowald (2005)) and simulated ensemble mean (middle) aeolian deposition of iron. Due to its large dynamic range, deposition flux is shown on a logarithmic scale. Deposition is in $\mu$mol Fe m$-2$ y$^{-1}$.

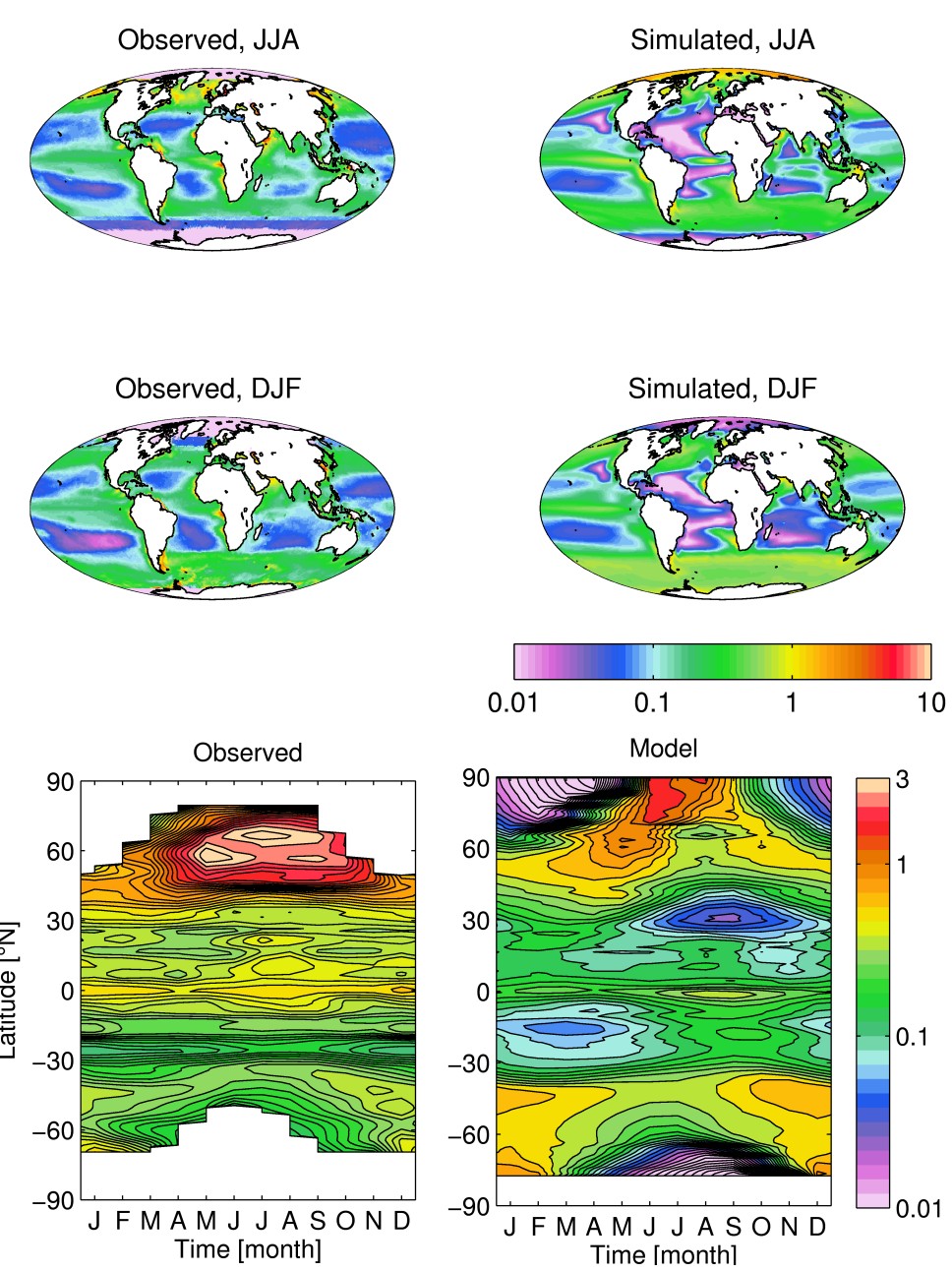

**Figure 13.** Observational (left; SeaWiFS) and simulated (right) surface chlorophyll, shown geographically for northern (top; JJA)and southern summer (middle; DJF), and as zonal Hovmöller diagrams (bottom). Missing observational data at high latitudes are shown as near-zero because of polar night / sea-ice. Concentrations in mg chl m$^{-3}$.

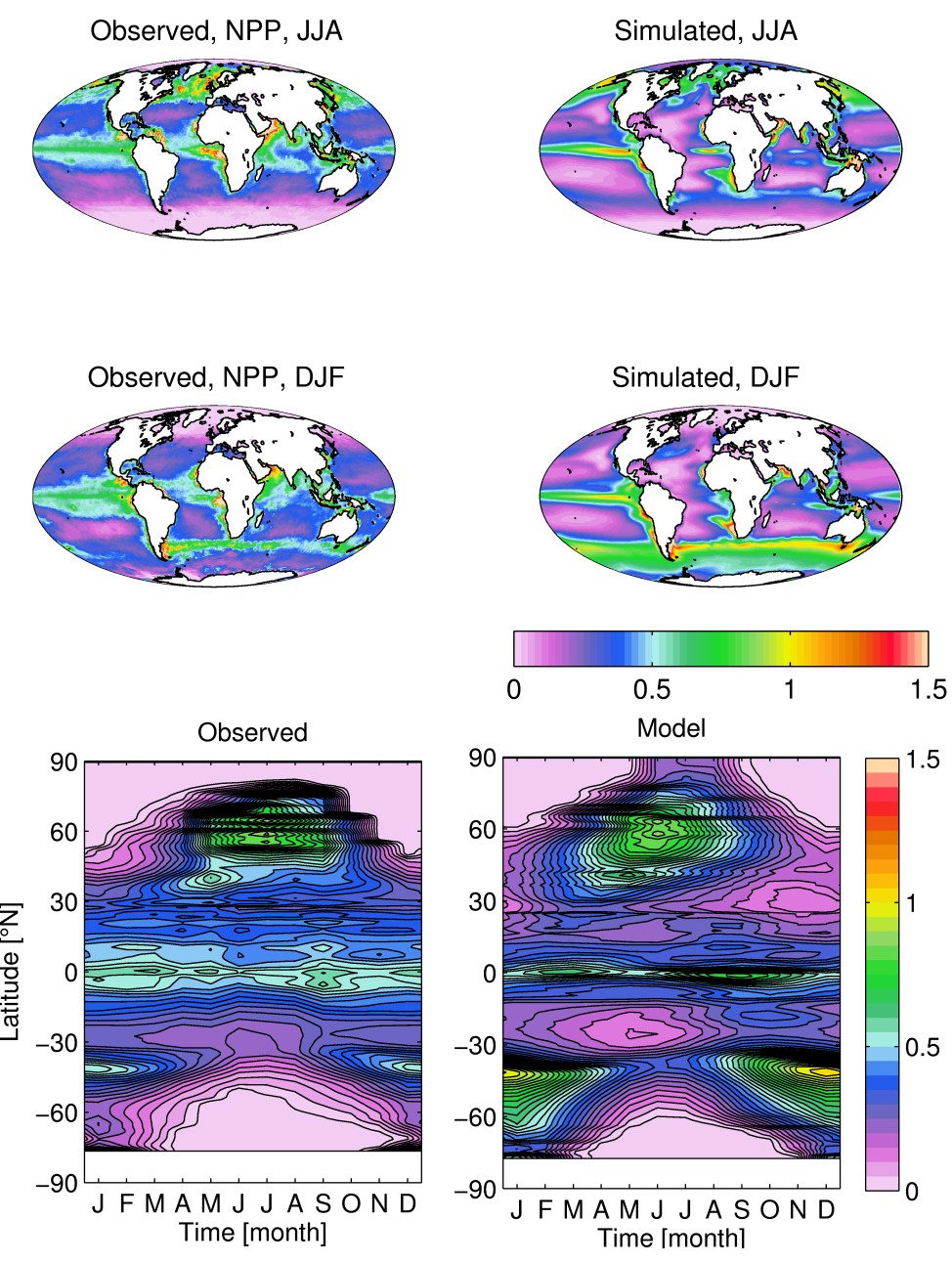

**Figure 14.** Observational (left) and simulated (right) vertically–integrated net primary production, shown geographically for northern (top; JJA)and southern summer (middle; DJF), and as zonal Hovmöller diagrams (bottom). Missing observational data at high latitudes are shown as zero because of polar night / sea-ice. Primary production in g C m$^{-2}$ d$^{-1}$.

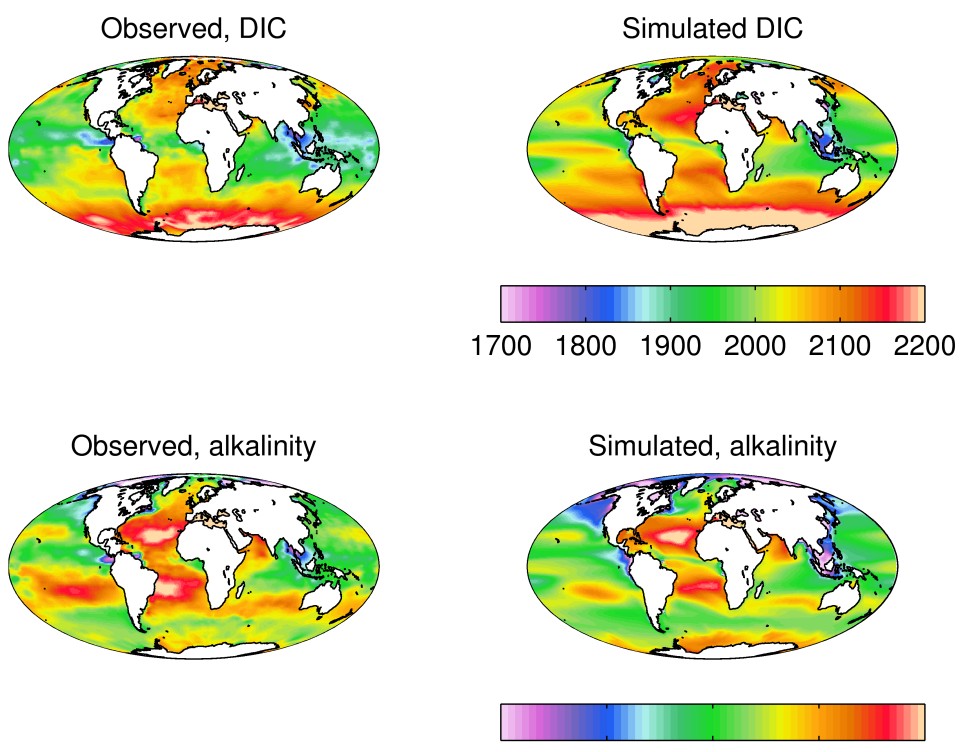

**Figure 15.** Observational (left; GLODAPv2) and simulated (right) annual average surface dissolved inorganic carbon (top) and total alkalinity (bottom). DIC in mmol C m$^{-3}$, alkalinity in meq m$^{-3}$.

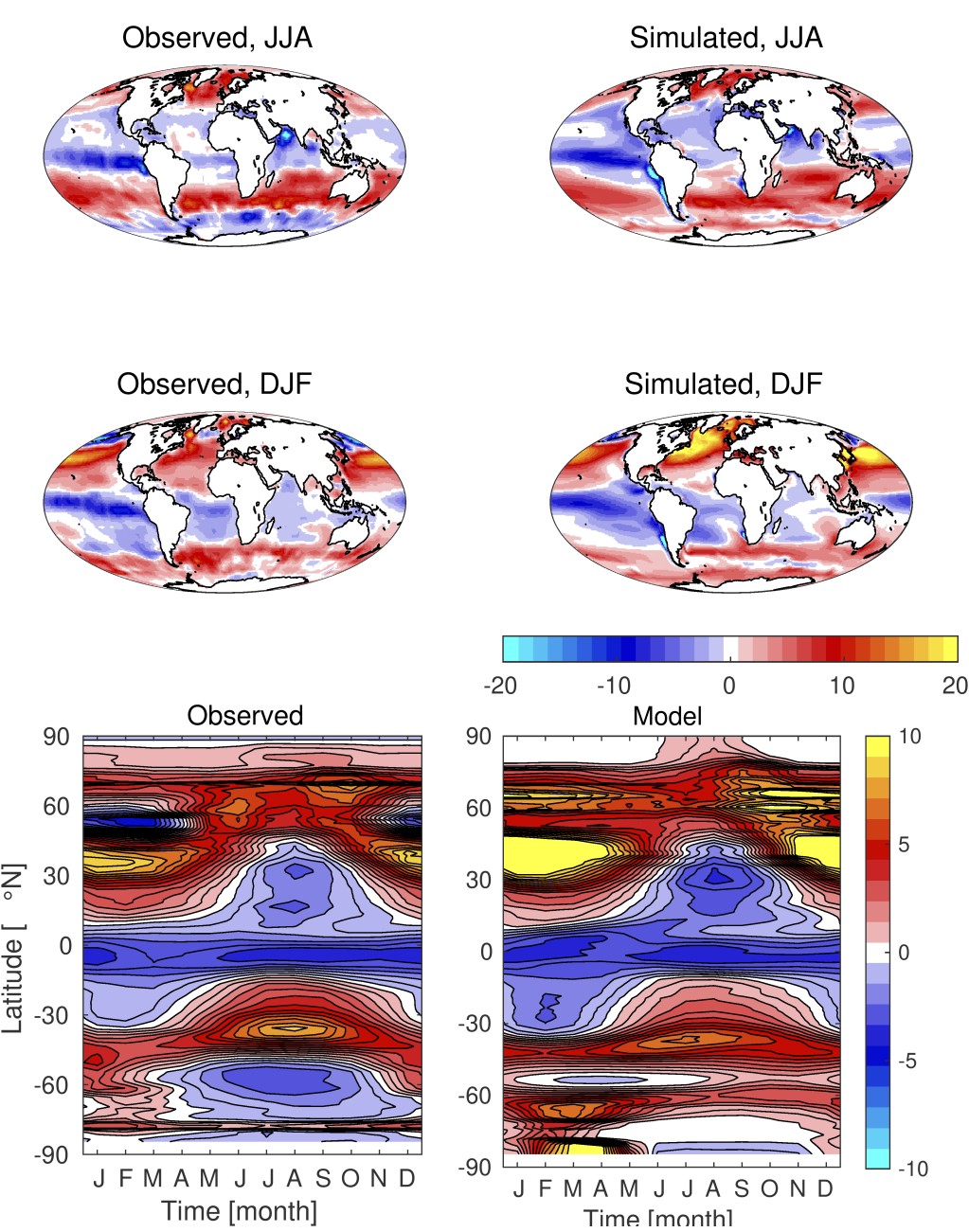

**Figure 16.** Observed (left; Rödenbeck et al. (2013)) and simulated (right) air-sea $CO_2$ flux, shown geographically for northern (top; JJA)and southern summer (middle; DJF), and as zonal Hovmöller diagrams (bottom). Red colours indicate $CO_2$ flux into the ocean, while blue colours denote outgassing $CO_2$. Flux in mmol C m$^{-2}$ d$^{-1}$.



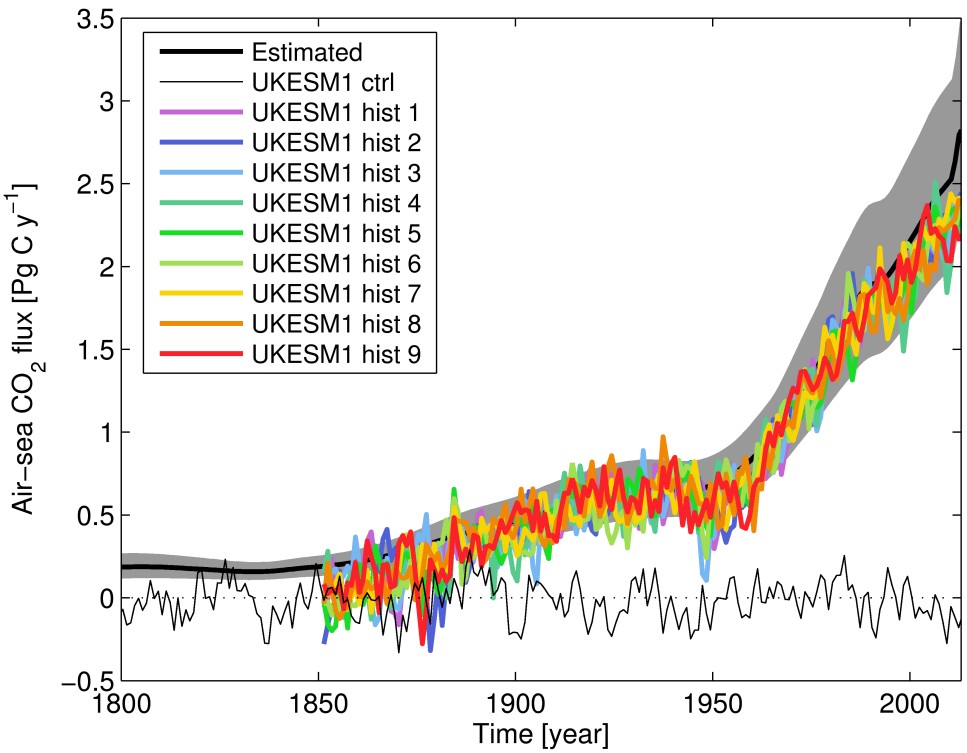

**Figure 17.** Time-series of globally integrated air-to-sea $CO_2$ flux, showing observationally-estimated mean (solid black) and range (grey shading; Khatiwala et al., 2009), and simulated control (dashed black) and historical (solid red) fluxes. Air-to-sea fluxes in Pg C $y^{-1}$. Note that while CMIP6 experiments begin in year 1850, the Industrial Revolution – and uptake of anthropogenic $CO_2$ by the ocean – began prior to this date.

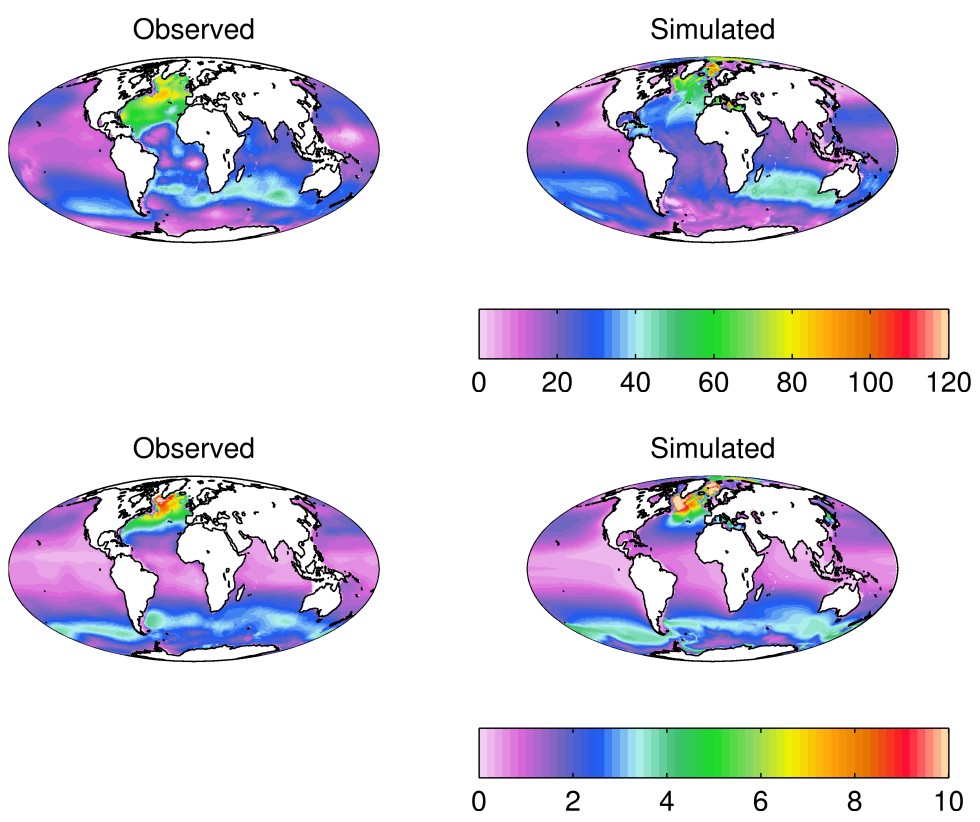

**Figure 18.** Observed (left; Key et al., 2004) and modelled (right) vertically-integrated anthropogenic $CO_2$ (top; mol C m$^{-2}$) and CFC-11 (bottom; $\mu$mol m$^{-2}$) in the 1990s. GLODAPv1.1 is used here as the time period used overlaps that of observational CFC-11.






**Figure 19.** Observed (GLODAPv1.1; Key et al., 2004) and modelled vertical profiles of DIC (top left; mmol C m$^{-3}$), alkalinity (top right; meq m$^{-3}$), anthropogenic CO$_2$ (bottom left; mmol C m$^{-3}$) and CFC-11 (bottom right; nmol m$^{-3}$). GLODAPv1.1 is used here as the time period used (the 1990s) overlaps that of observational CFC-11. Model profiles from the ensemble used in this analysis are presented individually, and are geographically masked according to GLODAPv1.1.

**Figure 20.** Observed (left; Key et al., 2004) and modelled (right) Atlantic sections (30°N) of anthropogenic $CO_2$ (top; mmol C m$^{-3}$) and CFC-11 (bottom; nmol m$^{-3}$) in the 1990s.



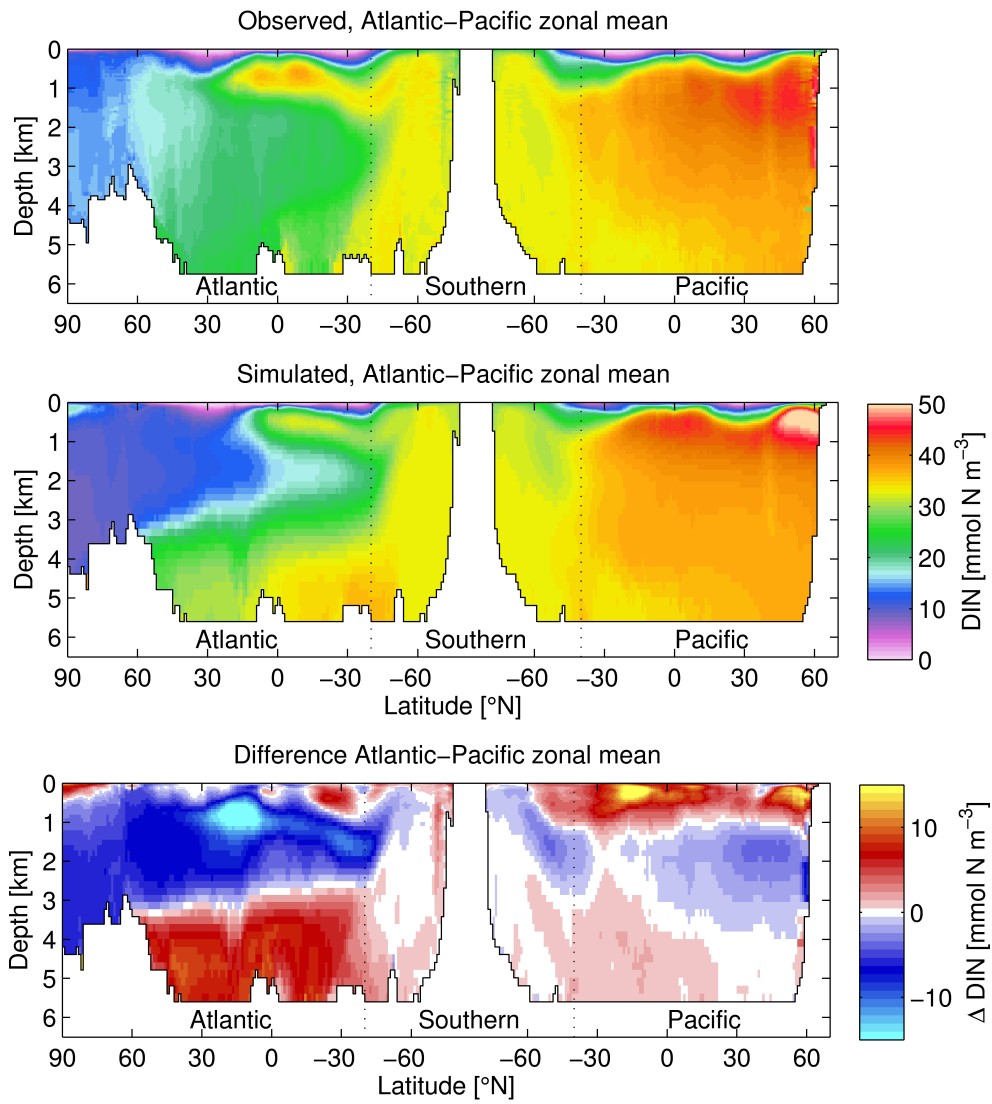

**Figure 21.** A "thermohaline circulation" section of observed (top) and modelled (bottom) zonal average dissolved inorganic nitrogen. Figure 6 explains the format of this section. Concentrations in mmol N m$^{-3}$.



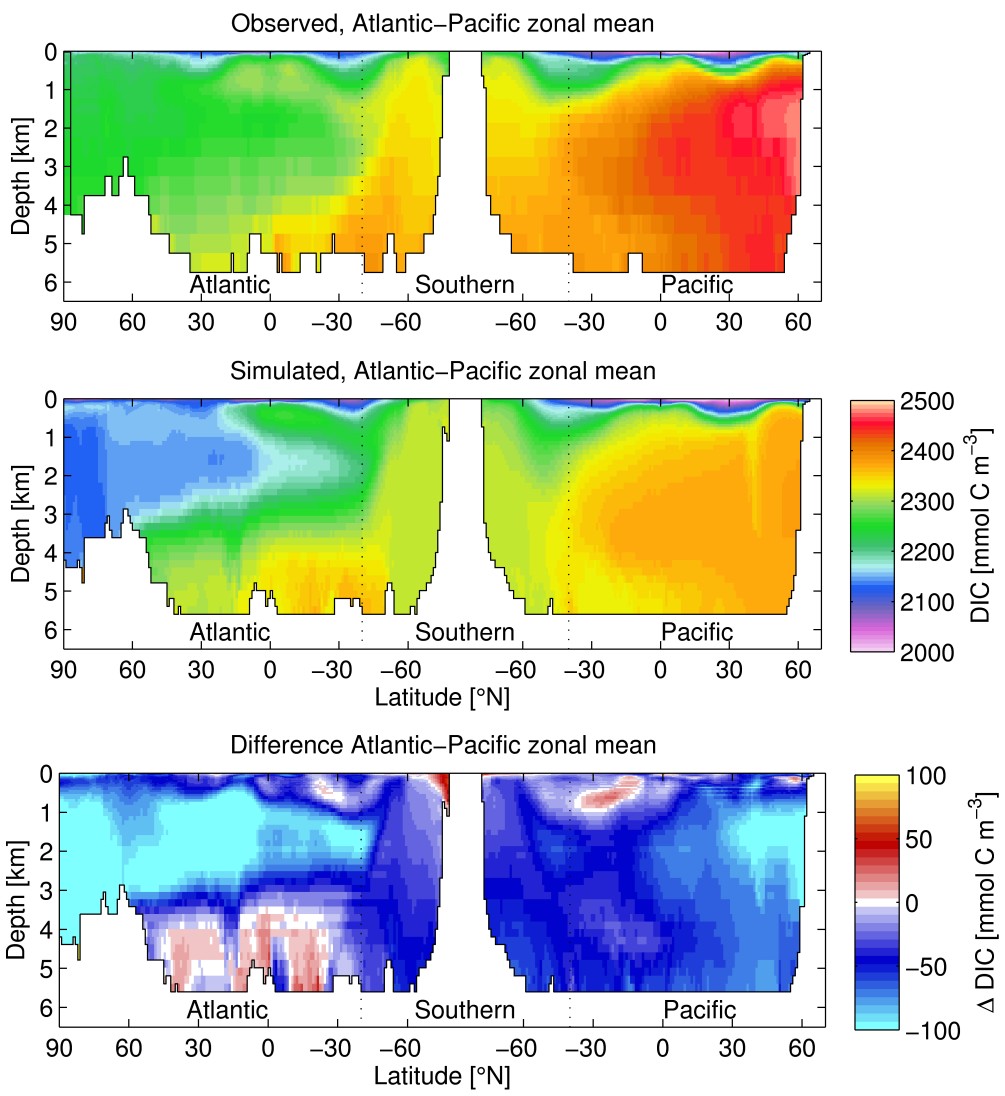

**Figure 22.** A "thermohaline circulation" section of observed (top) and modelled (bottom) zonal average dissolved inorganic carbon. Figure 6 explains the format of this section. Concentrations in mmol C m$^{-3}$.



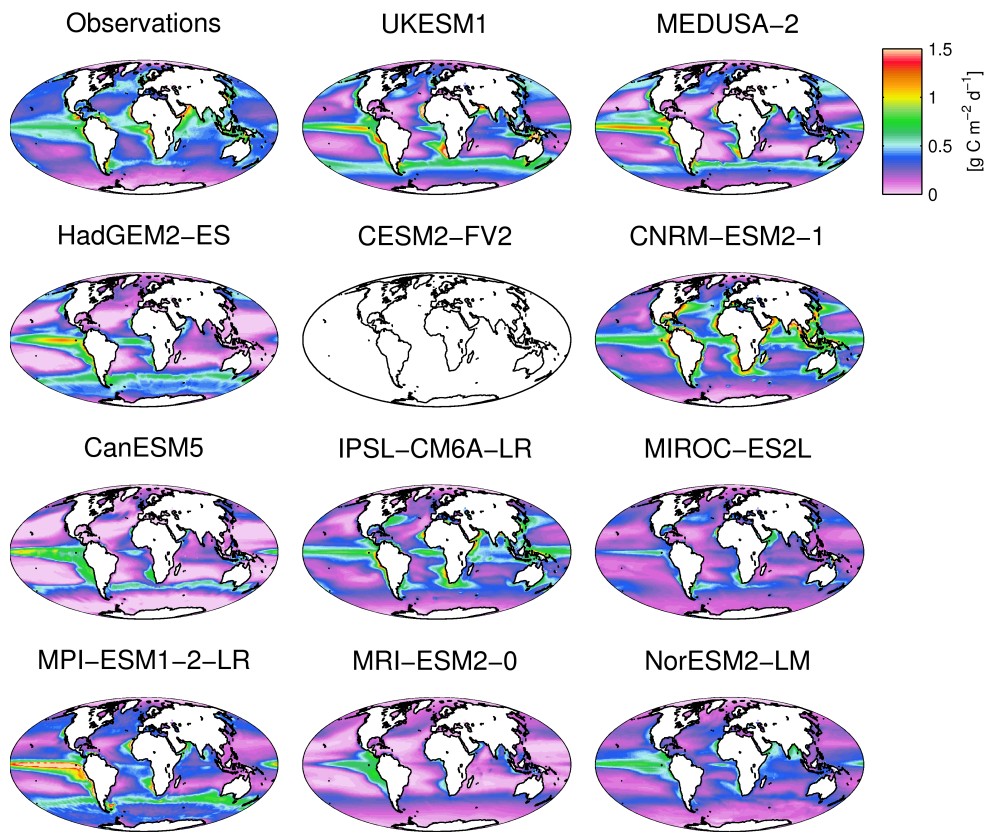

**Figure 23.** Intercomparison of annual mean vertically-integrated primary production between observed (top row, left), UKESM1 simulated (top row, centre) and a range of comparable CMIP6 models (rows 2–4). Results from CMIP5's precursor to UKESM1, HadGEM2-ES (Jones et al., 2011; row 2, left) and MEDUSA–2 (Yool et al., 2013; top row, right) are shown for comparison. Production in g C m$^{-2}$ d$^{-1}$.


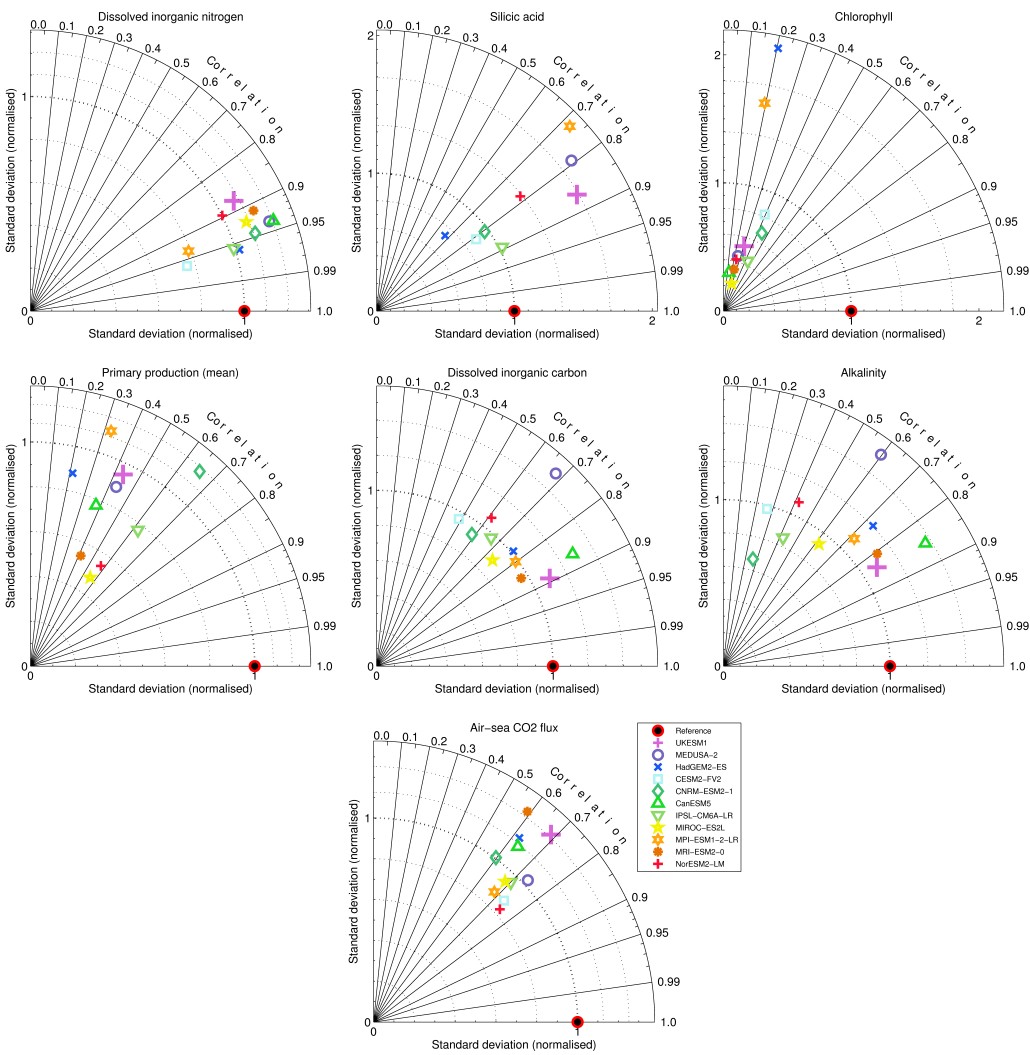

**Figure 24.** Taylor diagrams illustrating the skill of UKESM1 (and its precursors, HadGEM2-ES and MEDUSA-2) and a series of CMIP6 models over a set of standard surface ocean of biogeochemical properties: dissolved inorganic nitrogen (top left), silicic acid (top centre), chlorophyll (top right), primary production (middle left), dissolved inorganic carbon (middle centre), alkalinity (middle right), and air-sea $CO_2$ flux (bottom centre). The diagrams show model-observation comparisons based on annual average spatial fields, all regridded to the same standard grid. The diagrams share a common model key (bottom centre).



| Property | Mean | $\sigma$ | Min. | Max. | $M$ |
|---|---|---|---|---|---|
| AMOC | 15.83 | 1.191 | 12.60 | 18.80 | 0.200 |
| [Sv] | *14.77* | *0.877* | *12.61* | *16.88* | *-0.024* |
| Drake | 151.52 | 5.908 | 139.75 | 164.18 | -0.197 |
| [Sv] | *154.33* | *4.017* | *144.84* | *163.37* | *0.128* |
| SST | 17.76 | 0.175 | 17.45 | 18.42 | 0.015 |
| [°C] | *17.67* | *0.075* | *17.45* | *17.88* | *-0.002* |
| Temperature | 3.78 | 0.008 | 3.76 | 3.80 | -0.000 |
| [°C] | *3.77* | *0.007* | *3.76* | *3.79* | *-0.001* |
| SSS | 34.31 | 0.015 | 34.27 | 34.34 | -0.001 |
| [PSU] | *34.31* | *0.013* | *34.28* | *34.34* | *-0.001* |
| Salinity | 34.73 | 0.000 | 34.73 | 34.73 | -0.000 |
| [PSU] | *34.73* | *0.000* | *34.73* | *34.73* | *-0.000* |
| N sea-ice | 12.23 | 0.519 | 10.69 | 13.39 | -0.025 |
| [$10^6$ km$^2$] | *12.15* | *0.387* | *11.18* | *13.30* | *0.017* |
| S sea-ice | 11.35 | 0.890 | 8.34 | 13.08 | -0.092 |
| [$10^6$ km$^2$] | *11.87* | *0.554* | *10.44* | *13.27* | *-0.013* |
| MLD | 50.06 | 0.729 | 48.02 | 52.17 | 0.017 |
| [m] | *49.99* | *0.547* | *48.61* | *51.59* | *-0.010* |

**Table 1.** Selected ocean physical properties averaged across both the Historical ensemble (upper rows) and corresponding segments of the piControl (lower *italicised* rows). For each property, the statistics refer to the full 165 y period from 1850-2015. The final statistic, *M*, is the linear slope of the change in the property across this full period.





| Property [units] | Mean | $\sigma$ | Min. | Max. | $M$ |
|---|---|---|---|---|---|
| Surface DIN | 7.52 | 0.146 | 7.18 | 7.90 | 0.008 |
| [mmol N m$^{-3}$] | *7.49* | *0.125* | *7.20* | *7.82* | *0.003* |
| Surface silicic acid | 9.30 | 0.312 | 8.69 | 10.10 | 0.038 |
| [mmol Si m$^{-3}$] | *9.16* | *0.189* | *8.66* | *9.58* | *-0.010* |
| Surface iron | 0.52 | 0.005 | 0.50 | 0.53 | -0.000 |
| [$\mu$mol Fe m$^{-3}$] | *0.52* | *0.004* | *0.51* | *0.53* | *0.000* |
| Surface DIC | 2020.70 | 16.299 | 2000.76 | 2059.53 | 3.267 |
| [mmol C m$^{-3}$] | *2002.35* | *1.043* | *1999.81* | *2005.20* | *-0.017* |
| Surface alkalinity | 2317.76 | 1.617 | 2314.60 | 2321.42 | 0.265 |
| [meq m$^{-3}$] | *2316.09* | *0.817* | *2314.10* | *2317.93* | *-0.053* |
| Surface O$_2$ | 252.05 | 0.735 | 249.28 | 253.33 | -0.060 |
| [mmol O$_2$ m$^{-3}$] | *252.42* | *0.334* | *251.43* | *253.35* | *0.009* |
| Ocean O$_2$ | 190.50 | 0.373 | 189.87 | 191.17 | 0.050 |
| [mmol O$_2$ m$^{-3}$] | *190.41* | *0.279* | *189.88* | *190.89* | *0.051* |
| NPP | 47.96 | 0.728 | 46.04 | 49.83 | -0.014 |
| [Pg C y$^{-1}$] | *48.01* | *0.682* | *46.05* | *49.74* | *-0.003* |
| Air-sea CO$_2$ flux | 0.81 | 0.674 | -0.17 | 2.45 | 0.129 |
| [Pg C y$^{-1}$] | *-0.02* | *0.118* | *-0.33* | *0.26* | *-0.001* |
| Aeolian iron | 2.41 | 0.228 | 1.89 | 3.07 | -0.002 |
| [Gmol Fe y$^{-1}$] | *2.41* | *0.219* | *1.88* | *3.12* | *0.003* |

**Table 2.** Selected ocean biogeochemical properties averaged across both the Historical ensemble (upper rows) and corresponding segments of the piControl (lower *italicised* rows). For each property, the statistics refer to the full 165 y period from 1850-2015. The final statistic, *M*, is the linear slope of the change in the property across this full period.



| Field | Global | Southern | St S Atl | Eq Atl | St N Atl | Sp N Atl | St S Pac | Eq Pac | St N Pac | Sp N Pac | Arctic |
|---|---|---|---|---|---|---|---|---|---|---|---|
| *Surface DIN* | | | | | | | | | | | |
| Observed | 5.227 | 23.473 | 3.904 | 0.366 | 0.436 | 4.444 | 2.515 | 2.472 | 0.456 | 9.942 | 3.298 |
| Model | 7.757 | 25.676 | 4.952 | 0.058 | 0.273 | 3.039 | 8.543 | 8.004 | 3.256 | 9.632 | 8.503 |
| *Surface silicic acid* | | | | | | | | | | | |
| Observed | 7.512 | 32.019 | 2.665 | 2.048 | 1.487 | 3.185 | 1.855 | 2.828 | 3.151 | 21.299 | 8.329 |
| Model | 10.092 | 62.875 | 5.869 | 0.411 | 0.743 | 2.163 | 1.978 | 0.415 | 0.430 | 4.496 | 4.572 |
| *Surface chlorophyll* | | | | | | | | | | | |
| Observed | 0.219 | 0.164 | 0.249 | 0.361 | 0.195 | 0.517 | 0.131 | 0.184 | 0.143 | 0.558 | 0.342 |
| Model | 0.262 | 0.387 | 0.335 | 0.071 | 0.106 | 0.406 | 0.252 | 0.249 | 0.138 | 0.509 | 0.472 |
| *Primary production* | | | | | | | | | | | |
| Observed | 0.317 | 0.099 | 0.345 | 0.555 | 0.359 | 0.360 | 0.277 | 0.461 | 0.323 | 0.338 | 0.115 |
| Model | 0.356 | 0.309 | 0.466 | 0.324 | 0.217 | 0.365 | 0.358 | 0.510 | 0.263 | 0.466 | 0.153 |
| *Surface DIC* | | | | | | | | | | | |
| Observed | 2071 | 2192 | 2113 | 2036 | 2100 | 2117 | 2074 | 1991 | 2008 | 2058 | 2053 |
| Model | 2058 | 2211 | 2096 | 2022 | 2100 | 2084 | 2055 | 1981 | 1985 | 2030 | 2021 |
| *Surface Alkalinity* | | | | | | | | | | | |
| Observed | 2355 | 2350 | 2410 | 2386 | 2446 | 2353 | 2373 | 2317 | 2324 | 2268 | 2206 |
| Model | 2327 | 2361 | 2371 | 2363 | 2438 | 2299 | 2335 | 2291 | 2286 | 2201 | 2171 |
| *Air-sea $CO_2$ flux* | | | | | | | | | | | |
| Observed | 1.043 | -0.047 | 2.156 | -1.493 | 1.331 | 5.136 | 1.665 | -2.623 | 2.229 | 1.755 | 2.748 |
| Model | 1.350 | 2.122 | 1.862 | -1.556 | 0.573 | 8.050 | 1.171 | -3.442 | 1.954 | 5.120 | 3.176 |

**Table 3.** Selected biogeochemical properties averaged for specific geographical regions for annual mean fields. Observed and model values shown, with model values averaged over the Historical ensemble. Regional abbreviations are "St" for subtropical ($10°–40°$), "Eq" for equatorial ($10°S–10°N$) and "Sp" for subpolar ($40°–70°$). The Indian Ocean is excluded from this analysis for simplicity. Throughout, the model domain used matches that available from observational fields.