# Peer review of "Evaluating the physical and biogeochemical state of the global ocean component of UKESM1 in CMIP6 Historical simulations"

_Geoscientific Model Development, 2020_

## Referee Comment (RC1) · Anonymous Referee #1 · 25 Dec 2020

General Comments:

The authors are the developers of the Earth system model UKESM1, which is participating in CMIP6. In this study, they focused on the ocean physics and biogeochemical fields of the historical experiments using UKESM1, and analyzed the results by comparing them with various observations and other CMIP6 ESMs. They have done a comprehensive and good analysis of the UKESM1 performance, which I think provides useful information for those who are planning to do multi-model analysis using CMIP6 ESMs and for other ESM developers. I have the following questions and comments, which I hope will help improve the manuscript.

[Figure]

Specific Comments:

Line 140 "Evaluation uses the period 2000-2009 of the CMIP6 Historical simulation and compares to corresponding periods of observational data." The analysis mainly uses the results of simulations and observations for the period 2000-2009. This period is known for its negative IPO index, so the observations include that signal. On the other hand, since the model takes 9 ensemble members, the effect of internal variability is expected to be negligible. This may contribute to the equatorial Pacific warm bias of UKESM1 (see Section 3.1), although it may be small. If possible, it would be better to extend the time period used in the analysis to 20 or 30 years. Even if this is difficult, it would be good to take a longer-term look at SST to confirm that the model bias of UKESM1 does not depend on how the average period was taken.

Line 189 "In the Arctic, sea-ice is typically multi-year, and this positive bias in modelled area is accompanied by excessively thick sea-ice." I think that this sentence is confusing and needs to be rewritten.

Line 230 "This is more pronounced in the Atlantic basin, in particular at tropical latitudes, where midwater (100-1000 m) biases up to 4 C are found in the model" I think that this is influenced by the high temperature bias in the formation region of the NADW. If this is the case, why don't we use Figure 1 to discuss the temperature bias in the deep layers in relation to the formation process of the NADW?

Line 233 "These show the model ocean, particularly the Pacific basin, to be more stratified vertically compared to observations, with generally lower density surface waters (< 1000 m) overlying more dense deep waters." This may be due to inadequate and overall small parameterization of vertical mixing, the mechanism that transfers heat from the surface to the deeper layers. In fact, the bottom layer circulation is weakened accordingly (Figure 8). It would not be a bad idea to point out that the parameterization of vertical mixing is insufficient.

Line 249 "Strong sinking around Antarctica, combined with a slightly weaker NADW, is

consistent with the colder and fresher conditions shown for the deep ocean (particularly the Atlantic) in Figures 6 and 7," It is difficult to understand why "consistent" is used, so please rewrite it in more detail.

Line 409 "CFC-11" It is difficult to follow the discussion because of the sudden appearance of CFC-11 here; the description of CFC-11 in Figure 18 appears later in Line 421, so please reconsider the structure of this section.

Line 584 "the weaker deep overturning cell north of the ACC combined with a NADW cell which is slightly too weak (Figure 8) leads to a colder and fresher deep ocean" I would like to see a more detailed description of the mechanism of why the weaker deep overturning cell north of the ACC and the slightly too weak NADW cell lead to a colder and fresher deep ocean.

Section 4.2 I would like to see a detailed description of the improvements in the model from the predecessor HadGEM2-ES, and what could be fixed to make it better. The distribution of silicic acid concentration seems to have improved significantly. (Figure S22) And, before the detailed description of the figure (Figure 23) in the main text, there is a detailed description of the supplement figures (Figures S21-S23), which seems odd to me, so please reconsider the structure of this section.

Technical Corrections:

Line 6 "a new Earth system (ESM)" should be "a new Earth system model (ESM)".

Line 175 "% citepmarzocchi2015" typo.

Line 178 Typo. The umlaut is attached to the m.

Line 196 "Much as with sea-ice extent itself, UKESM1 performs best in the Arctic," I don't think "best" is the right word.

Line 295 "Table 2" I think this should be "Table 3". If we do that, then Table 2 will not be mentioned in the main text.

Line 382 "the SO" The abbreviation "SO" is not used elsewhere except Line 499, so it should not be used here as well.

Line 499 "SO" See the above comments on Line 382.

Line 727 "the other three cycles" C, Fe, and O2? It's difficult to tell what cycles you are referring to, so please specify C, Fe, and O2.

Line 759 "MOCSY-2.0 Orr and Epitalon (2015)" This should be "MOCSY-2.0 (Orr and Epitalon, 2015)"

Figures Please add labels to each panel in each figure.

Figure 1 It is easier to understand if observed, simulated, and differences are arranged in a single column or row.

Figure 2 A color palette should be created that is white at 0.15 or less.

Figure 4 It is easier to understand if observed, simulated, and differences are arranged in a single column or row. There is no explanation of the differences figures in the caption.

Figure 8 Contour interval should be noted in the caption.

Figure 12 m-2 should be m^{-2} (superscript).

Figures 13 and 14 These include Hovmoeller diagrams, but aren't they unnecessary? There is no detailed description of the monthly variation in the main text.

Figure 19 The first time it is mentioned in the main text is probably in Line 464. This is later than Figures 20, 21, and 22. The numbering of the figures should be in the order in which they are mentioned.

Tables Various values are listed in the tables, but in the main text, only AMOC in Table 1 (Line 268) and silicic acid in Table 2 (3?) (Line 295) seem to be mentioned. It is good to include various values in the tables, but the text should be revised so that it is not

assumed that only a small part of the tables is covered in the main text.

Supplement Figures The numbering of the supplement figures should be in the order in which they are mentioned in the main text. Please check.

Figure S3 There is no explanation of the differences figure in the caption.

Figure S4 It is easier to understand if observed, simulated, and differences are arranged in a single column or row. This figure is not mentioned in the main text? If not, please remove it.

Figure S6 "Potential density anomaly in kg m$-3$ (minus 1000 kg m$-3$)." This description is confusing and needs to be rewritten.

Figures S18 It would be good to add observed limiting nutrients (e.g., Moor et al., 2013, Table S2, DOI:10.1038/NGEO1765) for comparison.

———————————————————

---

## Referee Comment (RC2) · Anonymous Referee #2 · 18 Jan 2021

In this manuscript, Yool et al. present in detail the performance of the ocean and marine biogeochemical component of UKESM1, a novel Earth system models contributing to CMIP6. The manuscript is clearly written and provides a large array of standard analysis to understand the performance of UKESM1 at replicating observed features of the ocean and marine biogeochemical dynamics. This work presents an important basis for the users and other climate research groups in the context of multi-model analysis. I only have three majors and a set of minor comments that aims to clarify some point of the paper.

Major comments:

[Figure]

1- Although the authors did a great job assessing model modern climatology (2000-2009) against available modern observations, they didn't provide any key statistical metrics that might be useful to support their assessment (correlation, total root-mean squared errors, etc.). This would represent an added value in the current manuscript In addition, The abstract states that this paper investigate the driving mechanisms behind model-data errors. This is misleading. The manuscript remains largely speculative on what causes model biases and what mechanisms are at play to explain errors propagation or amplification. Within digging to much in those mechanisms (I reckon the paper is already long), I would like to see more properties-to-properties diagrams. This kind of analysis would strengthen the manuscript and support the conclusions. Those limited range of analysis mirrors the paper structure. Indeed there is no Methodology section. The reader remains without information on how model data are compared to observation (regridding for instance); how the mixed-layer depth are calculated; what is the working hypothesis to compute anthropogenic carbon and so on. This section, even short, would be useful.

2- Following major comment #1, I find it difficult to understand the choices of the authors team for the analysis. For instance, why DJF or JJA are taken for ocean analysis instead of JAS more widely accepted to studied the Southern Ocean winter dynamics. Does it has something to do with biases in the atmospheric model? Further explanation would be useful.

3- Manuscript structure: Although the manuscript is well written, some parts could be improved. For instance, I find it unexpected to find the description of the ocean and marine biogeochemical model in the appendix (while central in this work) but not in the main text to guide analyzes. Besides, the opening to the future scenarios seems out of the scope in the manuscript. This latter could be removed to give more space to develop key aspects (see point 1).

Minor comments: L6 observational properties = observed fiels ?

L15 compares favourably = you mean "outperforms" ? or just compares to other models ?

L29 in response to the release = driven by the release

L30 chemical composition = CO2 airborne fraction

L57 to identify avenues for future. . . addressing model limitation and weakness

L70 built to "simulate"

L80 do you simply mean that land-surface model is a submodule of the atmosphere model ?

L92 Mulcahy et al. (subm.) I don't know if it match GMD standard to refer to submitted papers (I counted two papers with this status)

L102-123 I would further develop this paragraph because it describes the model description central to this paper. In addition I suggest the authors to include the MEDUSA description here. Further details on how couplings may influence MEDUSA results (for instance, dust deposition).

L145 physics=hydrodynamics

L150 it is unclear what is meant with "ocean circulation", please provide further detail

L162: Please include Khatiwala et al. (2009) dataset

L175 "% citep. . ." I guess there is a typo here. Please check the following sentence

L215 "thermohaline transects" or "thermohaline circulation" Figures: this naming is misleading. What about "basin-averaged section" ?

L213 Section 3.2 please include detail on how to determine NADW and AABW properties in the model

L254: RAPID-MOCHA time coverage poorly matches with the time period chosen for

computing the model climatology. Does this difference in time period influences model assessment?

L295 "63 vs." ? please check

L317 biological community = marine biology ?

L334 three models = three algorithms ?

L348-362: All the Figures referred in these paragraphs should be included in the manuscript. Otherwise please consider moving this paragraph and associated text in the suppl. Mat.)

L596: Section 4.2: While interesting, I would suggest to stratify this section by comparing first NEMO-based model (IPSL, CNRM, CanESM) and then the other models. This would help to discuss the performance of the marine biogeochemical in a more constrained modelling framework. Please note that the Danabasoglu et al. and Voldoire et al. are inaccurate.

L681-695: Section 5: some key points would require further detail: resolution-dependent surface biases : is it assessed in this paper or based on published literature ?; same does for the aerosol-driven strengthening of the AMOC.

Figure 1: please detail somewhere in the ms why JJA and DJF has been used for seasonal analyzes

Figure 2: 0.15 isolines may help to compare model and observation

Figure 3: please considering remove indiv ens members and show $\pm 1$ standard deviation. Please explain what is behind seasonal minima/maxima.

Figure 6: please see the comments above

Figure 17: further discussion would be required to explain why model and obs-derived estimates differ over the preindustrial period (1800-1850) and why models fail at cap-

[Figure]

Interactive
comment

turing carbon uptake over the recent years (2000-2014). Please include the database in the reference list

Figure 18: please explain what is your working hypothesis to compute anthropogenic carbon

Figure 20: typo on the Figure "CO 2"

Figure 21 and 22: These two Figures merits further explanations. They show major model biases in the subsurface Atlantic waters whereas the modelled meridional transport matches well with the observed one.

technical suggestion: please consider to adjust color scales for some figures and use red-green color-blind color palette (where relevant)

---

## Author Comment (AC1) · 5 Mar 2021

In the following, referee comments are in black, while our responses are in green and added material is indicated in blue.

**Referee 1**

**General Comments:**

The authors are the developers of the Earth system model UKESM1, which is participating in CMIP6. In this study, they focused on the ocean physics and biogeochemical fields of the historical experiments using UKESM1, and analyzed the results by comparing them with various observations and other CMIP6 ESMs. They have done a comprehensive and good analysis of the UKESM1 performance, which I think provides useful information for those who are planning to do multi-model analysis using CMIP6 ESMs and for other ESM developers. I have the following questions and comments, which I hope will help improve the manuscript.

We would like to first thank the referee for their diligent reading of what's a rather long manuscript, made much longer by its supplementary material. Thank you!

**Specific Comments:**

Line 140 "Evaluation uses the period 2000-2009 of the CMIP6 Historical simulation and compares to corresponding periods of observational data." The analysis mainly uses the results of simulations and observations for the period 2000-2009. This period is known for its negative IPO index, so the observations include that signal. On the other hand, since the model takes 9 ensemble members, the effect of internal variability is expected to be negligible. This may contribute to the equatorial Pacific warm bias of UKESM1 (see Section 3.1), although it may be small. If possible, it would be better to extend the time period used in the analysis to 20 or 30 years. Even if this is difficult, it would be good to take a longer-term look at SST to confirm that the model bias of UKESM1 does not depend on how the average period was taken.

This is an excellent point. Our analysis focuses on an standardised recent period and does not consider the impact of variability, including important climatic modes. However, this point is worth mentioning, and we have added a further supplementary figure (S4) and the text below to address this.

"SST exhibits a number of major climate modes such as the Interdecadal Pacific Oscillation (IPO) and Atlantic Multidecadal Oscillation (AMO) that can introduce persistent and large-scale shifts in temperature that are of comparable magnitude to the model biases identified above. For instance, the IPO has a negative index (cooler than reference) during the time period shown in Figure 1, but a positive index (warmer than reference) during the preceding two decades (Salinger et al., 2001; Hu et al., 2018). Models also have climate modes, but these can be out of phase with those observed, and they may occlude or exaggerate biases. Supplementary Figure S4 partially addresses this by repeating the difference plot from Figure 1, but for the three preceding decades. The resulting patterns of model-observation difference are generally consistent between the decades and for both seasons, suggesting that they represent model biases rather than variability mismatch. In particular, persistent features include the strong cold bias in the western North Atlantic, warm biases in the equatorial Atlantic and Pacific basins (the latter seasonally), and a general warm bias in the Southern Ocean. As most other observational datasets used in the evaluation of UKESM1 properties are more restricted in the time periods they have available, similar analyses are more difficult. However, given the primary role of SST in many ocean processes, the apparent dominance of model bias in SST over its temporal

variability is suggestive that mismatches in major climatic modes is of secondary importance in our analysis."

Line 189 "In the Arctic, sea-ice is typically multi-year, and this positive bias in modelled area is accompanied by excessively thick sea-ice." I think that this sentence is confusing and needs to be rewritten.

We have amended this as follows:

"In the Arctic, sea-ice typically persists for multi-year periods, such that this bias towards excess ice area in UKESM1 is accompanied by sea-ice cover that is also excessively thick."

Line 230 "This is more pronounced in the Atlantic basin, in particular at tropical latitudes, where midwater (100-1000 m) biases up to 4 C are found in the model" I think that this is influenced by the high temperature bias in the formation region of the NADW. If this is the case, why don't we use Figure 1 to discuss the temperature bias in the deep layers in relation to the formation process of the NADW?

The largest temperature biases (4°C) are more associated with northward AAIW than southward NADW. This is perhaps clearer in the nitrogen nutrient figure where there is a distinct arm of high nutrient waters moving northward in the 100-1000 m region, and which appears associated with this positive temperature bias. Nonetheless, the warm bias in NADW appears associated with a corresponding bias in SST in the subpolar North Atlantic, so the following sentence has been added:

"The bias in southward-moving NADW (> 1000 m) is consistent with the warm bias in SST shown in its subpolar source regions in Figure 1."

Line 233 "These show the model ocean, particularly the Pacific basin, to be more stratified vertically compared to observations, with generally lower density surface waters (< 1000 m) overlying more dense deep waters." This may be due to inadequate and overall small parameterization of vertical mixing, the mechanism that transfers heat from the surface to the deeper layers. In fact, the bottom layer circulation is weakened accordingly (Figure 8). It would not be a bad idea to point out that the parameterization of vertical mixing is insufficient.

This clarification would be useful. We have added the following text to the end of this paragraph:

"This bias suggests that the model's parameterisation of vertical mixing may be insufficient, reducing the transfer of heat from the surface to deeper layers (and potentially weakening the deeper circulation; see below)."

Line 249 "Strong sinking around Antarctica, combined with a slightly weaker NADW, is consistent with the colder and fresher conditions shown for the deep ocean (particularly the Atlantic) in Figures 6 and 7," It is difficult to understand why "consistent" is used, so please rewrite it in more detail.

We would agree that the text is unclear on its use of "consistent". We have amended as below to more clearly note that the deep biases mentioned are consistent with a more dominant role for AABW in UKESM1:

"Stronger sinking in UKESM1 around Antarctica, combined with a slightly weaker NADW than observed, indicates a more dominant role for AABW in the model, and is consistent with the colder and fresher biases found in the deep ocean (particularly the Atlantic) in Figures 6 and 7, as well as biases in biogeochemical fields (see later)."

Line 409 "CFC-11" It is difficult to follow the discussion because of the sudden appearance of CFC-11 here; the description of CFC-11 in Figure 18 appears later in Line 421, so please reconsider the structure of this section.

We agree that this is an omission. We have added the following text to the model description section to clarify CFC-11's role:

"In addition to the biogeochemical tracers of MEDUSA-2.1, UKESM1 includes the chlorofluorocarbon tracer, CFC-11. This artificial tracer has an atmospheric time-history analogous to that of anthropogenic $CO_2$, and can be used as a marker for recently ventilated watermasses (Key et al, 2004). It can be measured from seawater samples with high accuracy, and provides an additional measure here for evaluating simulated circulation."

Line 584 "the weaker deep overturning cell north of the ACC combined with a NADW cell which is slightly too weak (Figure 8) leads to a colder and fresher deep ocean" I would like to see a more detailed description of the mechanism of why the weaker deep overturning cell north of the ACC and the slightly too weak NADW cell lead to a colder and fresher deep ocean.

Yes, the reference to NADW is confusing here. We have rewritten this to reflect the clearer point that we intended to make, namely that the sluggish AABW cell in the Atlantic contributes to its positive nutrient and negative oxygen biases:

"For instance, the weak deep overturning AABW cell north of the ACC (Figure 8) reduces the ventilation rate of the abyssal Atlantic, and contributes to the build-up of nutrients and the corresponding depletion of oxygen."

Section 4.2 I would like to see a detailed description of the improvements in the model from the predecessor HadGEM2-ES, and what could be fixed to make it better. The distribution of silicic acid concentration seems to have improved significantly. (Figure S22)

We would agree that this is an omission in the manuscript. The relationship between HadGEM2-ES and UKESM1, and in particular their ocean components, is not made clear in the text. To address this, we have added the paragraph below to the model description section:

"UKESM1 is the successor model to its CMIP5 predecessor, HadGEM2-ES (Collins et al., 2011). Many of its components are evolved versions of those in the earlier model, including its land surface, physical atmospheric core, and atmospheric chemistry components (Sellar et al., 2019). However, in the specific case of the ocean in UKESM1, its dynamical core, grid domain, sea-ice, and marine biogeochemistry are wholly new and replace the corresponding components in HadGEM2-ES. Consequently, there is no direct traceability between the oceans of the two generations of CMIP model. Nonetheless, as part of the assessment of UKESM1, elements of its performance relative to that of HadGEM2-ES are examined in Section 4.2."

And, before the detailed description of the figure (Figure 23) in the main text, there is a detailed description of the supplement figures (Figures S21-S23), which seems odd to me, so please reconsider the structure of this section.

To avoid overcrowding the main body of the manuscript with figures, we decided to select a single example from the set of primary biogeochemical properties examined. These properties are introduced and discussed in the same order as earlier in the manuscript (hence the figure ordering), but we elected to choose primary production for the main body because of its high diversity in model representation.

**Technical Corrections:**

Line 6 "a new Earth system (ESM)" should be "a new Earth system model (ESM)".

Amended as suggested.

Line 175 "% citepmarzocchi2015" typo.

Thanks for spotting this. Corrected.

Line 178 Typo. The umlaut is attached to the m.

Thanks for spotting this. Corrected.

Line 196 "Much as with sea-ice extent itself, UKESM1 performs best in the Arctic," I don't think "best" is the right word.

We would agree. Changed "best" for "better".

Line 295 "Table 2" I think this should be "Table 3". If we do that, then Table 2 will not be mentioned in the main text.

Thanks for spotting this. We have corrected this and have expanded the description of Tables 2 and 3.

Line 382 "the SO" The abbreviation "SO" is not used elsewhere except Line 499, so it should not be used here as well.

Thanks. Abbreviation removed.

Line 499 "SO" See the above comments on Line 382.

As above.

Line 727 "the other three cycles" C, Fe, and O2? It's difficult to tell what cycles you are referring to, so please specify C, Fe, and O2.

Yes, this is correct. We have amended the text to:

"The model's nitrogen, silicon and alkalinity cycles are closed and conservative (e.g. no riverine inputs), while the cycles of iron, carbon and oxygen are open."

Line 759 "MOCSY-2.0 Orr and Epitalon (2015)" This should be "MOCSY-2.0 (Orr and Epitalon, 2015)"

Thanks. Amended.

**Figures**

Please add labels to each panel in each figure.

We tried this, but couldn't get a consistent "finish" to the resulting Figure panels due to figures being a mix of single and multiple panels. However, we have implemented other suggestions around figures, and hope these make up for this omission. From the captions, it should be clear in all cases which panels are which.

Figure 1 It is easier to understand if observed, simulated, and differences are arranged in a single column or row.

We've rearranged figures so that observations, model and their delta are arrange in single columns corresponding to the two seasons shown. Other figures which don't have a delta have also been altered for consistency.

Figure 2 A color palette should be created that is white at 0.15 or less.

This is a good suggestion – we have amended the figure so that the colourscale shows white for values < 0.15.

Figure 4 It is easier to understand if observed, simulated, and differences are arranged in a single column or row. There is no explanation of the differences figures in the caption.

See previous response.

Figure 8 Contour interval should be noted in the caption.

Contours are every 2 Sv. This has been added to the caption as directed.

Figure 12 m-2 should be m^{-2} (superscript).

Fixed - thanks!

Figures 13 and 14 These include Hovmoeller diagrams, but aren't they unnecessary? There is no detailed description of the monthly variation in the main text.

The Hovmoller diagrams are not referred to specifically, but the description of both figures in the main text does discuss the seasonality of the properties, and the associated model biases. As such we have retained them. Also, the chlorophyll plot is not very flattering to our model, and deleting it feels like we'd be hiding poor results.

Figure 19 The first time it is mentioned in the main text is probably in Line 464. This is later than Figures 20, 21, and 22. The numbering of the figures should be in the order in which they are mentioned.

Thanks for spotting this. The mention of the figure got moved in manuscript drafting, but the figure itself did not. We have amended this.

**Tables**

Various values are listed in the tables, but in the main text, only AMOC in Table 1 (Line 268) and silicic acid in Table 2 (3?) (Line 295) seem to be mentioned. It is good to include various values in the tables, but the text should be revised so that it is not assumed that only a small part of the tables is covered in the main text.

We have added a few more references to Tables 1-2 in the main text so that readers are more clearly directed to the variability and trends that they report. Including:

"Table 1 lists the global means (or mean integrals) of these surface physical properties across both the full Historical period and the corresponding piControl period. For both of these simulation ensembles, the variability and ranges of each of these properties are given, together with the simple linear trend over the full 165 y period."

"The influx of $CO_2$ into the surface ocean is also documented in Table 2's mean and trend statistics of surface DIC and air-sea flux, in particular how they compare with the corresponding piControl period."

Supplement Figures The numbering of the supplement figures should be in the order in which they are mentioned in the main text. Please check.

As well as Supplementary Figures being referred to out of order, we found two figures that were no longer mentioned in the text. These have been removed, and the order of Supplementary Figures has been amended to follow the order in which they are mentioned in the main text.

Figure S3 There is no explanation of the differences figure in the caption.

We have added reference to this in the caption.

Figure S4 It is easier to understand if observed, simulated, and differences are arranged in a single column or row. This figure is not mentioned in the main text? If not, please remove it.

Thanks for spotting this omission. Figure corrected and missing text on SSS added to the main text:

"Supplementary Figure S4 parallels Figure 1, showing the observed (WOA, 2013) and simulated sea surface salinity (SSS) for summer and winter, together with (model - observed) differences. UKESM1 shows a general negative bias in SSS (~1 PSU), but with significant regions of positive bias in the tropical Atlantic and Indian oceans (< 1 PSU). There are also "hotspots" of bias in the Bay of Bengal (positive), off the west (negative) and east (positive) coastline of equatorial South America, in the Yellow and East China seas (negative), and in the Arctic (both positive and negative). These regions are mostly located close to major riverine inputs, and likely reflect model inaccuracies in the precise location and magnitude of associated freshwater additions."

Figure S6 "Potential density anomaly in kg m  3 (minus 1000 kg m  3)." This description is confusing and needs to be rewritten.

We would agree. We have reworded this as:

"Potential density is shown as kg m-3 minus 1000 kg m-3 (i.e. the actual density range in the upper panels is 1025 to 1028 kg m-3)."

Figures S18 It would be good to add observed limiting nutrients (e.g., Moor et al., 2013, Table S2, DOI:10.1038/NGEO1765) for comparison.

We have amended to include the following text around nutrient limitation. However, the lack of biogeochemical detail in our model precludes a thorough comparison, and we have noted this.

"Corresponding observational patterns of nutrient stress are more sparsely available (Moore et al., 2013). However, UKESM1's nutrient limitation overlaps the major observed patterns, including widespread nitrogen stress in the Atlantic Ocean, and iron stress throughout the Pacific and Southern oceans, as well as at high latitudes in the North Atlantic (Moore et al., 2013). Nonetheless, the simplicity of MEDUSA prevents it from representing the limitation of phytoplankton found by Moore et al. (2013) for the macronutrient, phosphorus, and the micronutrients, cobalt, zinc and vitamin B12."

---

## Author Comment (AC2) · 5 Mar 2021

In the following, referee comments are in black, while our responses are in green and added material is indicated in blue.

**Referee 2**

In this manuscript, Yool et al. present in detail the performance of the ocean and marine biogeochemical component of UKESM1, a novel Earth system models contributing to CMIP6. The manuscript is clearly written and provides a large array of standard analysis to understand the performance of UKESM1 at replicating observed features of the ocean and marine biogeochemical dynamics. This work presents an important basis for the users and other climate research groups in the context of multi-model analysis. I only have three majors and a set of minor comments that aims to clarify some point of the paper.

Again, we would like to first thank the referee for their careful reading of what is a very long manuscript, made much longer by its supplementary material. Thank you!

**Major comments:**

1- Although the authors did a great job assessing model modern climatology (2000-2009) against available modern observations, they didn't provide any key statistical metrics that might be useful to support their assessment (correlation, total root-mean squared errors, etc.). This would represent an added value in the current manuscript In addition, The abstract states that this paper investigate the driving mechanisms behind model-data errors. This is misleading. The manuscript remains largely speculative on what causes model biases and what mechanisms are at play to explain errors propagation or amplification. Within digging to much in those mechanisms (I reckon the paper is already long), I would like to see more properties-to-properties diagrams. This kind of analysis would strengthen the manuscript and support the conclusions.

While we appreciate that our analysis is incomplete, we are uncertain as to the sort of additional analysis that the referee would like to see (e.g. specific property-property plots). We are happy to act on further guidance here. Given the already long nature of this paper, we likely prefer to defer a detailed analysis of the root cause of different biases to subsequent papers.

Those limited range of analysis mirrors the paper structure. Indeed there is no Methodology section. The reader remains without information on how model data are compared to observation (regridding for instance); how the mixed-layer depth are calculated; what is the working hypothesis to compute anthropogenic carbon and so on. This section, even short, would be useful.

We have addressed these omissions by adding the following paragraphs to the "Datasets and Evaluation" subsection of the manuscript's Methods section.

"In addition, several derived variables are calculated from observational and model fields.

– Mixed layer depth (MLD) is calculated in the same way from both observed and modelled 3D fields of potential temperature. MLD is determined to be the depth at which the vertical profile of potential temperature is 0.5C lower than that at the depth of 5 m. Alternative MLD schemes using similar thresholds in potential density (either fixed or variable with temperature) were also examined, but global coverage was less complete with these (especially in sea-ice regions), so the potential temperature criterion was favoured.

– Modelled integrated AMOC and Drake Passage transports are calculated here using the BGC-val toolkit (de Mora et al., 2018). In the case of AMOC, the calculations are based on those of Kuhlbrodt et al. (2007) and McCarthy et al. (2015) and use the cross-sectional area at the 26N transect to calculate the maximum depth-integrated current. Drake Passage transport is calculated following Donohoe et al. (2016) as the total, depth-integrated current along a north-south transect between the South American continent and the Antarctic Peninsula. The methods for both transports are described in de Mora et al. (2018).

– Model anthropogenic $CO_2$ is estimated by differencing DIC fields from the Historical simulation of each ensemble member with the corresponding DIC field from the piControl at the same relative timepoint. For example, we estimate anthropogenic $CO_2$ in 1990 from a given Historical ensemble member as the difference between this member's DIC field at this particular time and the DIC field from the piControl simulation from the same timepoint, i.e. the time that corresponds to 140 years (i.e. 1990 - 1850 = 140) after the Historical ensemble member branched from the piControl. This approach aims to account for drift in the simulations, although it omits changes driven by divergence in circulation and biogeochemistry between the Historical and piControl simulations. These are assumed to be small in this method."

"Throughout, fields of observational and model properties are plotted on their original horizontal and vertical grids. Where these properties are directly intercompared, for instance in difference plots, observational fields are first regridded to the model grid (using the scatteredInterpolant function of Matlab v2020a). In Section 4.3, horizontal fields of UKESM1 output are compared with those from fellow CMIP6 models, and here all models are regridded to a common, uniform 1° grid."

2- Following major comment #1, I find it difficult to understand the choices of the authors team for the analysis. For instance, why DJF or JJA are taken for ocean analysis instead of JAS more widely accepted to studied the Southern Ocean winter dynamics. Does it has something to do with biases in the atmospheric model? Further explanation would be useful.

The choice of DJF and JJA is simply following meteorological conventions for "winter" and "summer". We have added the following text into the Model Evaluation section:

"A number of figures illustrate observed and modelled properties (and the biases of the latter) for the June-July-August (JJA) and December-January-February (DJF) meteorological seasons that correspond respectively to northern hemisphere summer and winter (and southern hemisphere winter and summer)."

3- Manuscript structure: Although the manuscript is well written, some parts could be improved. For instance, I find it unexpected to find the description of the ocean and marine biogeochemical model in the appendix (while central in this work) but not in the main text to guide analyzes. Besides, the opening to the future scenarios seems out of the scope in the manuscript. This latter could be removed to give more space to develop key aspects (see point 1).

Both the ocean physical and biogeochemical submodels are outlined in the main body. While the ocean physical model is documented more completely elsewhere (and this is noted via cited papers), the precise version of the biogeochemical model used here is not, hence the appendix sections. However, we have retained it as supplementary so as not to "break the flow" of the manuscript by adding a large block of text descriptive of a single submodel.

**Minor comments:**

L6 observational properties = observed fiels ?

Yes, this is more accurate. Amended.

L15 compares favourably = you mean "outperforms" ? or just compares to other models?

This is a fair point. The model's actual performance is mixed, so outperforms sometimes (e.g. surface DIC, ALK), underperforms in others (e.g. surface DIN). Overall, "favourably" tried to capture this, but not well. We have rewritten this to:

"performs well alongside its fellow members of the CMIP6 ensemble"

L29 in response to the release = driven by the release

More accurate. Amended as suggested.

L30 chemical composition = CO2 airborne fraction

This is better.  Amended as suggested.

L57 to identify avenues for future: : : addressing model limitation and weakness

Yes, this was rather clunky. We have reworded to:

"Third, to identify avenues for addressing model limitations and weaknesses in future versions"

L70 built to "simulate"

This is better. Amended.

L80 do you simply mean that land-surface model is a submodule of the atmosphere model ?

The suggested phrasing downplays the actual separation of the land and atmosphere submodules (they can be run independently). To address the confusion, we have reworded to:

"In outline, UKESM1 is comprised of closely-coupled atmosphere and land submodules that are linked through an explicit coupler module, OASIS3-MCT_3.0, to coupled ocean and sea-ice submodules."

L92 Mulcahy et al. (subm.) I don't know if it match GMD standard to refer to submitted papers (I counted two papers with this status)

One of the three submitted manuscripts is now accepted, and its entry in the bibliography has been updated to reflect this. The two others are in late stages of pre-publication and we anticipate publication ahead of this manuscript. should this not occur, we will revise to replace with appropriate alternative publications.

L102-123 I would further develop this paragraph because it describes the model description central to this paper. In addition I suggest the authors to include the MEDUSA description here. Further details on how couplings may influence MEDUSA results (for instance, dust deposition).

Per our previous remarks, we have decided to retain the description of MEDUSA within the appendix. However, the referee's suggestion that we identify the specific couplings between MEDUSA and the rest of the model is good, and we have added the following text:

"Within UKESM1, MEDUSA interacts with other model components via the following feedback connections: atmosphere-ocean exchange of $CO_2$; ocean-to-atmosphere fluxes of DMS and PMOA; deposition of terrestrial iron to the ocean via atmospheric dust transport."

L145 physics=hydrodynamics

We have amended "physics" to "physical" - T and S (the variables in question) are more state than hydrodynamics.

L150 it is unclear what is meant with "ocean circulation", please provide further detail

ECCO essentially estimates the patterns and magnitudes of ocean circulation. For clarity, we have altered this to: "hydrodynamic circulation state".

L162: Please include Khatiwala et al. (2009) dataset

We will add this dataset to the final version of the Zenodo archive that accompanies this manuscript.

L175 "% citep: : :" I guess there is a typo here. Please check the following sentence

Thank you. Amended.

L215 "thermohaline transects" or "thermohaline circulation" Figures: this naming is misleading. What about "basin-averaged section" ?

We have retained the description, but have added the following to make it clear how the figures have been created

"These transects are created from basin zonal means of the plotted properties."

L213 Section 3.2 please include detail on how to determine NADW and AABW properties in the model

We have not determined NADW and AABW properties directly during our analysis. Rather, the model's meridional overturning circulation (MOC) is calculated by integrating its velocity field in the same manner as the ECCO product. As the watermass structure is very similar between UKESM1 and ECCO, we use the same terminology, although our analysis does not go as far as formally identifying different watermasses. We do not judge that this level of in-depth analysis is necessary at this point.

L254: RAPID-MOCHA time coverage poorly matches with the time period chosen for computing the model climatology. Does this difference in time period influences model assessment?

The ensemble mean ranges 16-17 Sv in the period of RAPID-MOCHA (with variability 15-18 Sv). This lies within the range (14.6-19.3 Sv) observed at the array. To make the comparison clearer, we have added the RAPID-MOCHA time-series to the appropriate panel of Figure 9, and altered the text appropriately. Because the RAPID-MOCHA era is short compared to the full Historical period, we have also added Supplementary Figure S8 that focuses in on this period.

L295 "63 vs." ? please check

The values for Southern Ocean silicic acid quoted in the manuscript are correct. The model estimates both higher maximum values, and a much larger area of high concentrations within the Southern Ocean. A compounding factor in the numbers appearing questionable is a mistake in the description of the regions. This has been amended to more clearly describe the Southern Ocean.

"In the southern hemisphere, the subpolar region falls primarily within the Southern Ocean, although as its northern margin is delineated at -50N rather than -40N, the southern margins of the southern subtropical Atlantic and Pacific extend to -50N."

L317 biological community = marine biology ?

Yes, this is clearer. Amended.

L334 three models = three algorithms ?

Yes, they are much simpler models than the dynamic models within ESMs so "algorithm" is more accurate. Amended.

L348-362: All the Figures referred in these paragraphs should be included in the manuscript. Otherwise please consider moving this paragraph and associated text in the suppl. Mat.)

We agree. While other supplementary figures augment the description of properties which already feature in main body figures, these two figures do not. We have moved them into the main body accordingly and amended the text to reflect this.

L596: Section 4.2: While interesting, I would suggest to stratify this section by comparing first NEMO-based model (IPSL, CNRM, CanESM) and then the other models. This would help to discuss the

performance of the marine biogeochemical in a more constrained modelling framework. Please note that the Danabasoglu et al. and Voldoire et al. are inaccurate.

The aim of this section is simply to place the performance of UKESM1 within the context of fellow CMIP6 models. Identifying and attributing common patterns of bias is beyond the scope of this manuscript (but see Seferian et al. 2020; cited here). In any case, the behaviour of these three models is not completely convergent in the plots shown, and we judge that it would not be straightforward (or brief) to attempt to identify parallels. However, the referee's point around shared components is worth emphasising, and we have added the following text to this section:

"While the full configurations of these models are diverse, the CNRM-ESM2-1, CanESM5, and IPSL-CM6A-LR models share a common NEMO physical ocean with UKESM1, though they diverge on other components, including marine biogeochemistry."

We have also tried to amend the references to the correct ones.

L681-695: Section 5: some key points would require further detail: resolution dependent surface biases : is it assessed in this paper or based on published literature ?; same does for the aerosol-driven strengthening of the AMOC.

Both points are made elsewhere in the manuscript, and supported by published analyses. See 187-189 for resolution dependence, and 299-303 on aerosol forcing. They have not been separately evaluated here.

Figure 1: please detail somewhere in the ms why JJA and DJF has been used for seasonal analyses

See earlier comments.

Figure 2: 0.15 isolines may help to compare model and observation

Following a suggestion by another referee, the figure has been modified so that model values below 0.15 are omitted (the colour bar is altered to reflect this). This permits easier comparison of the model and observations. (A similar modification is made for the sea-ice thickness supplementary figure.)

Figure 3: please considering remove indiv ens members and show 1 standard deviation. Please explain what is behind seasonal minima/maxima.

We would agree that the figure gets rather dense from 1980 onwards when the observational data is also plotted. We have followed the referee's suggestion to amend this.

Regarding the seasonal minima and maxima, these are simply the months in which sea-ice usually records its minimum and maximum extents. We do not follow the question.

Figure 6: please see the comments above

See earlier comments.

Figure 17: further discussion would be required to explain why model and obs-derived estimates differ over the preindustrial period (1800-1850) and why models fail at capturing carbon uptake over the recent years (2000-2014). Please include the database in the reference list

On the first point, CMIP6 starts its Historical era in 1850, approximately a century after the industrial revolution began. The Khatiwala et al. (2009) product estimates $CO_2$ uptake over the more complete period during which atmospheric $pCO_2$ has been elevated by anthropogenic emissions.

On the second point, UKESM is typically in the lower portion of the uncertainty range of the Khatiwala et al. (2009) dataset throughout the Historical era. The seeming greater divergence at the very end of the simulated period may be exaggerated by a sudden uptake spike in the dataset, which itself is an estimate. However, to resolve this would require a detailed analysis of factors in the model (e.g. SST, SSS, wind speed) that we judge is beyond the scope of this manuscript.

Nonetheless, the description of UKESM1's comparison with Khatiwala et al. (2009) does not note that it falls within the lower portion of the uncertainty range throughout the Historical period. The text has been amended so that this is clear to readers.

 "With some variability, particularly in the early decades, the ensemble tracks the observationally-estimated uptake, reproducing the same pace and features, but with the ensemble estimating a slightly lower flux than estimated (88.5%; integrated 1850-2013)."

"The plot also shows UKESM1's piContol simulation to illustrate the magnitude and period of variability with constant background atmospheric $CO_2$. This shows CMIP6's Historical period beginning (and the piControl period ending) in 1850, approximately a century after the industrial revolution and significant fossil fuel $CO_2$ emissions began. This differs from the Khatiwala et al. (2009) product, which estimates ocean $CO_2$ uptake over the more complete period of anthropogenic emissions."

Finally, we will include the Khatiwala et al. (2009) dataset within the Zenodo archive that accompanies this manuscript.

Figure 18: please explain what is your working hypothesis to compute anthropogenic carbon

Historical ensemble members branch off from the piControl at different time points in its evolution. In a Historical simulation, the 1990s occur 140-149 years after this branch point. So, to estimate anthropogenic DIC concentration, the 1990s decade of each Historical ensemble member are differenced from the piControl decade 140-149 years after the branching point of this ensemble member. Other alternative approaches are possible, but we chose this approach here to account for drift in the piControl.

While lines 399-400 already describe the approach used, the description is very short and may be misunderstood. To avoid this, we have added the following text to the Methods section:

"Model anthropogenic $CO_2$ is estimated by differencing DIC fields from the Historical simulation of each ensemble member with the corresponding DIC field from the piControl at the same relative timepoint. For example, to estimate anthropogenic $CO_2$ in 1990 from a particular Historical ensemble member, we subtract the piControl state from the timepoint that corresponds to 140 years (i.e. 1990 - 1850 = 140) after the Historical ensemble member branched from it."

Figure 20: typo on the Figure "CO 2"

Amended.

Figure 21 and 22: These two Figures merits further explanations. They show major model biases in the subsurface Atlantic waters whereas the modelled meridional transport matches well with the observed one.

Modelled transport in UKESM1 is reasonable, although it is clear from a several properties (salinity, DIN, oxygen) that AABW is more pronounced in UKESM1. It is also clear that the AAIW limb of fresher water from the Southern Ocean is less pronounced than observed, transporting less nutrient to the North Atlantic.

The balance of watermasses - and their constituents - in UKESM1 suggests that nutrients are more efficiently "trapped" in AABW, decreasing the concentrations in upper watermasses and contributing to the pattern observed.

in the case of DIC, the biases match those of DIN, but with an additional negative bias imparted by reduced surface DIC concentrations. This is already described in the manuscript as a consequence of a negative surface alkalinity bias.

The manuscript has been modified around the distributions of DIN as follows:

"Meanwhile, in deeper waters the bias is reversed to strong positive, as the more sluggish AABW circulation shown in Figure 8 is ventilated less efficiently, accumulating excess DIN while accruing an oxygen deficit (Supplementary Figure S16)."

**Technical suggestion:**

please consider to adjust color scales for some figures and use red-green color-blind color palette (where relevant)

This is a fair point. To address this, we have adopted the perceptually-neutral Google "Turbo" palette where appropriate, and have retained the blue-white-red palette for "delta" plots. The "Turbo" palette is described, and its perceptual qualities evaluated, here ...

  https://ai.googleblog.com/2019/08/turbo-improved-rainbow-colormap-for.html

A notice about the palette is included in the acknowledgements to assist others wishing to use it.